EMBO
Molecular Medicine

# Salivary glands regenerate after radiation injury through SOX2-mediated secretory cell replacement

Elaine Emmerson[1],[†],[§] , Alison J May[1],[†], Lionel Berthoin[1], Noel Cruz-Pacheco[1], Sara Nathan[1], Aaron J Mattingly[1], Jolie L Chang[2], William R Ryan[2], Aaron D Tward[2] & Sarah M Knox[1],[*]

## Abstract

Salivary gland acinar cells are routinely destroyed during radiation treatment for head and neck cancer that results in a lifetime of hyposalivation and co-morbidities. A potential regenerative strategy for replacing injured tissue is the reactivation of endogenous stem cells by targeted therapeutics. However, the identity of these cells, whether they are capable of regenerating the tissue, and the mechanisms by which they are regulated are unknown. Using *in vivo* and *ex vivo* models, in combination with genetic lineage tracing and human tissue, we discover a SOX2+ stem cell population essential to acinar cell maintenance that is capable of replenishing acini after radiation. Furthermore, we show that acinar cell replacement is nerve dependent and that addition of a muscarinic mimetic is sufficient to drive regeneration. Moreover, we show that SOX2 is diminished in irradiated human salivary gland, along with parasympathetic nerves, suggesting that tissue degeneration is due to loss of progenitors and their regulators. Thus, we establish a new paradigm that salivary glands can regenerate after genotoxic shock and do so through a SOX2 nerve-dependent mechanism.

**Keywords** radiotherapy; regeneration; salivary gland; SOX2; stem cells
**Subject Categories** Regenerative Medicine; Stem Cells

## Introduction

Therapeutic radiation continues to be a life-saving treatment for cancer patients and is utilized for a spectrum of malignancies including those of the head and neck. Indeed, the vast majority of patients suffering head and neck cancer will receive radiotherapy in addition to chemotherapy and surgery (~60,000 new patients per year in US; Siegel *et al*, 2015). Although this combination treatment is highly efficacious in eliminating tumors, a severe side effect is damage and/or destruction of healthy tissue lying in the field of radiation. Such organs include the salivary glands, which exhibit tissue dysfunction even after low doses of radiation (Grundmann *et al*, 2009). At the higher doses routinely given to patients (60 Gy), off-target radiation destroys saliva-synthesizing acinar cells (Sullivan *et al*, 2005; Redman, 2008) and results in a lifetime of dry mouth and co-morbidities (e.g., tooth decay, oral infections, poor wound healing (Brown *et al*, 1975; Dreizen *et al*, 1977; Dusek *et al*, 1996). Although there has been success with intensity modulated radiation to spare one of the three major salivary glands (parotid), the proximity of the glands to the tumor sites often prevents application of this technique, leaving 80% of head and neck cancer patients with dry mouth syndrome (Lee & Le, 2008).

As with all other organs damaged by radiation, including the lungs, heart, and bladder (Emami *et al*, 1991), there are few, if any, treatments available to improve or restore tissue function. Current treatment options for cancer survivors suffering radiation-induced salivary dysfunction and degeneration focus on short-term relief from the symptoms, but no long-term restorative therapies are available. Regenerative strategies such as reactivating endogenous stem cells or transplanting non-irradiated stem cells have been proposed (Lombaert *et al*, 2008; Ogawa *et al*, 2013; Pringle *et al*, 2016). However, these applications are curtailed by the dearth of knowledge regarding the identity of adult salivary progenitor cells that contribute to acini under homeostatic or injury conditions. Although it was recently proposed that acinar cells are derived through self-duplication rather than from defined progenitors (Aure *et al*, 2015), an analysis of subpopulations of these cells for progenitor-like activity was not performed. It also remains to be determined whether acinar cells, either through self-duplication or through progenitor cell expansion, are capable of repopulating cells after genotoxic damage. Although a plethora of studies have utilized irradiated salivary glands as a model of degeneration (Zeilstra *et al*, 2000; Coppes *et al*, 2001, 2002), the regenerative capacity of adult salivary cells damaged by radiation has not been investigated *in vivo*.

How acinar cells are replaced during salivary gland homeostasis is also poorly understood. Studies in adult organs over the last

1 Program in Craniofacial Biology, Department of Cell and Tissue Biology, University of California, San Francisco, CA, USA
2 Department of Otolaryngology, University of California, San Francisco, CA, USA
*Corresponding author. Tel: +1 415 502 0811; E-mail: sarah.knox@ucsf.edu
†These authors contributed equally to this work as first authors
§Present address: The MRC Centre for Regenerative Medicine, The University of Edinburgh, Edinburgh, UK

150 years have clearly shown that peripheral nerves are essential for the maintenance of organ and tissue integrity (Erb, 1868). Skeletal muscle atrophies in the absence of stimulation by motor neurons (Fu & Gordon, 1995; Batt & Bain, 2013) and epithelial organs such as fungiform taste buds (Von Vintschgau & Honigschmied, 1877), prostate (Wang et al, 1991; Lujan et al, 1998) and the salivary gland degenerate after ablation of sensory and/or autonomic nerves (Schneyer & Hall, 1967; Mandour et al, 1977; Kang et al, 2010). Although it is unclear how nerves control tissue homeostasis for these organs, studies in skin indicate sensory nerves, through sonic hedgehog secretion, promote the self-renewal of adult epithelial stem cells and consequently the maintenance of the downstream cell lineage, that is, dome cells in the skin (Peterson et al, 2015; Xiao et al, 2015). In addition, studies in the salamander (Wallace, 1972) and embryonic salivary gland (Knox et al, 2013) suggest that peripheral nerves have the capacity to regenerate tissue via activation of multipotent stem cells, but evidence for this in the adult mammalian system is lacking.

Using a combination of mouse genetics, ex vivo cultures, and human tissue explants, we unexpectedly discover that salivary acini are capable of regenerating after radiation and do so in response to cholinergic activation through a progenitor cell-dependent mechanism. We show that SOX2 marks the sole progenitor for the acinar lineage that can replace acinar cells during homeostasis and after radiation-induced injury, indicating that salivary progenitors can withstand, at least in the short term, genotoxic shock. Importantly, treatment of healthy and irradiated tissue with cholinergic mimetics stimulated acinar cell replenishment. Thus, our data reveal the extensive regenerative capacity of the tissue even under genotoxic shock and suggest that targeting of SOX2$^+$ cells might be a therapeutic approach to regenerate tissue damaged by radiation therapy.

# Results

## SOX2 marks a progenitor cell that gives rise to acinar but not duct cells during salivary gland homeostasis

SOX2 has been established as a progenitor cell marker in the fetal mouse submandibular and sublingual salivary glands, but whether SOX2$^+$ cells in the adult tissue also produce acinar and duct cells is unclear (Arnold et al, 2011; Emmerson et al, 2017). Furthermore, whether these cells are also present in adult human salivary glands is not known. We found SOX2 to be expressed by a subset of acinar cells in all three of the major adult human salivary glands [Fig 1A, submandibular gland (SMG), sublingual gland (SLG), parotid gland (PG)]. In the mouse, SOX2 protein was restricted to the adult murine SLG (absent from the SMG and PG, Figs 1B and EV1A) where it was expressed by undifferentiated aquaporin (AQP)5-positive, mucin (MUC)19-negative acinar cells (21 ± 4% of all AQP5$^+$ acinar cells; Fig 1C and D). Consistent with their potential role as a progenitor cell, ~6% of SOX2$^+$AQP5$^+$ cells co-expressed Ki67 (Figs 1E and EV1B) while 19 ± 4% were in the cell cycle (CyclinD1$^+$; Fig EV1C). To determine whether SOX2$^+$ cells contributed to acinar and duct lineages, we performed genetic lineage tracing using Sox2$^{CreERT2}$ mice (Arnold et al, 2011) crossed to a Rosa26$^{mTmG}$ reporter strain. The Rosa26$^{mTmG}$ mouse is a double-fluorescent

reporter which when crossed with a Cre line expresses membrane-targeted tandem dimer Tomato (mT) prior to Cre-mediated excision and membrane-targeted green fluorescent protein (mG) after excision (Muzumdar et al, 2007; Fig EV1D). As such, lineage-traced cells will express mG. As shown in Fig 1F, SOX2$^+$ cells self-renew and produce differentiated acinar cells marked by AQP5 and MUC19 but not KRT8$^+$ duct cells after 14 or 30 days (Figs 1F and EV1E). Thus, our lineage tracing results indicate that SOX2$^+$ cells are lineage-restricted progenitor cells that give rise to differentiated progeny, similar to what has been observed in the epidermis, intestine, and incisor (Owens & Watt, 2003; Barker, 2014; Seidel et al, 2017).

Given KIT$^+$ cells, which reside primarily in the intercalated ducts of the SLG and SMG (Andreadis et al, 2006; Nelson et al, 2013), have previously been proposed to give rise to acinar cells in adult tissue (Lombaert et al, 2008; Nanduri et al, 2013, 2014; Pringle et al, 2016), we genetically traced these cells using the Kit$^{CreERT2}$ promoter crossed to the Rosa26$^{mTmG}$ reporter at 6 weeks of age. However, no KIT$^+$ cell-derived acinar cells (i.e., double positive for AQP5 and mG) were evident in either the SLG or SMG at 14 days or 6 months after induction (Fig EV1F). Instead, KIT$^+$ cells contributed exclusively to the intercalated ducts in the SLG (as can be observed by co-staining for the intercalated duct marker KRT8) and intercalated and larger ducts in the SMG. Thus, these data indicate that KIT$^+$ cells are progenitors for the ductal and SOX2$^+$ cells for the acinar lineage.

## SOX2 and SOX2$^+$ cells are essential for production of secretory acini

Our lineage tracing analysis confirmed that SOX2$^+$ cells give rise to acinar but not duct cells. However, as we also observed the presence of Ki67$^+$SOX2$^-$ acinar cells (~6% SOX2$^+$Ki67$^+$ and 16.5% SOX2-Ki67$^+$ cells, Fig EV1B), suggestive of an alternative progenitor cell or a transit-amplifying cell for the acinar lineage, we investigated the requirement of SOX2 and SOX2$^+$ cells in SLG maintenance and repair by genetically removing Sox2 in SOX2$^+$ cells using Sox2$^{CreERT2}$; Sox2$^{fl/fl}$ mice (Fig 2A and C) or ablating SOX2$^+$ cells using diphtheria toxin (DTA) expressed under the control of the inducible Sox2 promoter (Sox2$^{CreERT2}$;Rosa26$^{DTA}$; Fig 2B and D). In the latter assay, SOX2$^+$ cells undergo cell death in response to intracellular production of DTA. Ablation of Sox2 from SOX2$^+$ cells or elimination of SOX2$^+$ cells via DTA severely depleted SOX2$^+$ and AQP5$^+$ cells but not KRT8$^+$ ductal cells indicating Sox2 and SOX2$^+$ cells were necessary for maintaining functional acini (Fig 2A–D; efficiency of Sox2 or SOX2$^+$ cell ablation is shown in Fig EV2A). In the absence of Sox2, acinar but not ductal cells exited the cell cycle, as shown by the decrease in cyclin D1 (CCND1)$^+$ acinar cells (Fig EV2D; arrowheads indicate CCND1$^+$ cells and dotted white lines highlight ductal cells). Furthermore, ablation of SOX2$^+$ cells resulted in few remaining acini by 8 days (Figs 2B and D, and EV2A), as shown by large regions of the ductal network completely devoid of AQP5$^+$ cells (ducts are marked by dashed lines or KRT8 in Fig 2B). To exclude the possibility that tissue degeneration was solely due to destabilization of the tissue rather than loss of acinar cell replacement, we examined SLG after a short-term ablation. As shown in Appendix Fig S1, at day 4 or 5 (3 or 4 days of tamoxifen treatment), few SOX2$^+$ cells remained in the gland of both the

**Figure 1. SOX2 marks a progenitor cell that gives rise to acinar but not duct cells in the adult salivary gland.**

A   Representative image of adult human submandibular (SMG), sublingual (SLG), and parotid (PG) salivary gland (non-IR, 28–33 years) immunostained for SOX2, epithelia (E-cadherin; ECAD) or CD44, and nuclei. Single arrows indicate SOX2 expressing acinar cells. Scale bar is 20 μm.

B   Wild-type murine SMG and SLG stained for SOX2, ECAD, and nuclei. Arrowheads indicate SOX2 expressing cells. Scale bar is 50 μm. Yellow dashed line denotes border between SMG and SLG.

C   $Sox2^{eGFP}$ sublingual salivary glands (SLG) were immunostained for GFP and differentiated acinar marker mucin 19 (MUC19). White dashed lines outline $Sox2^{eGFP+}$ MUC19(−) cells. Scale bar = 20 μm.

D   AQP5$^+$SOX2$^+$ cells as a percentage of total AQP5$^+$ acinar cells.

E   SLG immunostained for SOX2, Ki67, and epithelial marker E-cadherin (ECAD). White arrow indicates proliferating Ki67$^+$SOX2$^+$ cell. White lines outline individual cells and nuclei. Scale bar = 10 μm.

F   Representative images of $Sox2$ lineage-traced SLG. Recombination was induced in $Sox2^{CreERT2};Rosa26^{mTmG}$ mice and salivary gland traced for 24 h and 30 days before immunostaining for SOX2, acinar markers AQP5 and MUC19, and ductal marker KRT8. * indicates MUC19(−) $Sox2^{CreERT2}$GFP(+) cells. Scale bars = 30 μm. mT = membrane-bound Tomato.

Data information: Cells quantified in (D) were counted from three non-consecutive sections of $n = 5$ female adult SLGs. Data are presented as mean ± SD.

$Sox2^{CreERT2}$; $Sox2^{fl/fl}$ and $Sox2^{CreERT2};Rosa26^{DTA}$ SLG (Appendix Fig S1A and B) and $Sox2$ transcripts were substantially reduced (Appendix Fig S1B). However, acini were present albeit disorganized and atrophic in appearance. Furthermore, we did not observe an increase in SOX2$^+$ cells (or $Sox2$ transcripts), indicating that SOX2 is not ectopically expressed in acinar cells in response to

tissue damage. We also determined whether alterations in tissue composition were due to reduced innervation, an essential regulator of tissue function. However, we measured similar innervation in $Sox2^{CreERT2}$; $Sox2^{fl/fl}$ SLG to wild-type controls and a significant increase in axon bundles in $Sox2^{CreERT2}$; $Rosa26^{DTA}$ SLG (Fig EV2B and C). The latter finding suggests ablation of cells triggers the release of factors that promote innervation but that, even with increased innervation, regeneration is not possible without SOX2$^+$ cells. In sum, these results indicate that SOX2$^+$ cells, at least under the conditions tested, are the sole acinar progenitors in the SLG and that acini do not arise from the self-duplication of fully differentiated acinar cells, as suggested previously (Aure et al, 2015). Similar to studies in the epidermis, intestine, and incisor (Owens & Watt, 2003; Barker, 2014; Seidel et al, 2017), our data also suggest the presence of a transit-amplifying population derived from SOX2$^+$ cells that may be involved in rapidly repopulating the acinar compartment.

## Parasympathetic nerves preserve SOX2$^+$ progenitors and promote SOX2-mediated acinar cell replacement

Adult murine and human salivary glands atrophy after removal of parasympathetic activity. However, the effect of denervation on acinar cell replacement and progenitor cells has not been investigated (Garrett et al, 1999; Raz et al, 2013). To this end, we denervated one of the two pairs of murine SLGs by transecting the chorda tympani (Fig 3A; contralateral glands were used as internal controls). After 7 days, transcript levels of neuronal genes Tubb3, Vip, and Vacht (Fig 3B, red bars) and GFRα2$^+$ or TUBB3$^+$ nerves (Figs 3C and EV3C) were severely reduced, indicating successful denervation. We did not observe a concurrent loss of the cholinergic muscarinic receptors Chrm1 and Chrm3 transcripts (Fig 3B, red bars); however, it is possible that in the absence of parasympathetic innervation a compensatory mechanism may maintain Chrm1 and Chrm3 transcription. Although the SLG is predominately served by the parasympathetic branch with very little sympathetic innervation in comparison (Emmelin et al, 1965), we did observe a reduction in sympathetic innervation following chorda tympani transection (Fig EV3A). As such, although the levels of sympathetic nerves are minor, we cannot rule out that some of the effects of denervation may be due to a loss of sympathetic input.

Similar to the effect of radiation therapy on tissue structure (Sullivan et al, 2005; Redman, 2008) adult acinar cells, as well as SOX2$^+$ progenitors, were more sensitive to the loss of innervation than ducts. Denervation resulted in reduced acinar cell size (as observed previously; Patterson et al, 1975; Fig EV3B) decreased AQP5 protein and transcript levels of the differentiated acinar cell marker Muc19 (Fig 3B and D). Interestingly, transcript and protein levels of MIST1 were unchanged following denervation (Fig 3B, E and F), suggesting that while functional markers of acinar cells are disrupted in the absence of innervation, acinar cell identity is not adversely affected. Strikingly, SOX2$^+$ cells lose expression of Sox2 (demonstrated using the Sox2eGFP mouse) and the levels of SOX2 protein and transcript were greatly reduced (Figs 3B, C and F, and EV3C), indicating SOX2 maintenance requires innervation. To determine whether SOX2$^+$ cells remained capable of repopulating the tissue after denervation, we performed genetic lineage tracing where

Cre driven by the endogenous Sox2 promoter ($Sox2^{CreERT2}$; $Rosa26^{mTmG}$) was activated 3 days after denervation and traced until day 14. As shown in Fig 3G and H, acinar cell replacement by SOX2$^+$ progenitors was significantly reduced (~50%) 14 days after transection. Similarly, in SLG in which recombination was induced before nerve transection (tamoxifen 1 day prior to transection), acinar cell replacement by SOX2$^+$ progenitors was significantly depleted (~50%) after 14 days (Fig EV3E and F). Reduced acinar cell replacement is likely due to decreased cell proliferation rather than cell death as we measured a reduction in Ccnd1, while markers of cell death [activated caspase-3 (CASP3$^+$) cells in Fig EV3G or Bax, Pmaip1 (NOXA) and Bbc3 (PUMA) in Fig EV3H] were either not observed or remained unchanged. The absence of cell death also suggests that cells that were previously positive for SOX2 continue to be present but that cholinergic innervation is essential for maintaining SOX2 expression.

To confirm that denervation preferentially affects the acinar lineage in the SLG, we also analyzed other epithelial cell lineages. KRT8$^+$ ducts in denervated glands resembled innervated controls, as did transcript levels of the ductal genes Krt7, Krt8, and Krt19 (Fig 3B, D and F). Furthermore, KRT5$^+$ cells, progenitors in developing SMG/SLG (Knox et al, 2010; Lombaert et al, 2013) that are maintained by parasympathetic nerves (Knox et al, 2010), were unaffected by denervation (Figs 3F and EV3D; for transcript Krt5 expression see Fig 3B). Based on these findings, we conclude that parasympathetic innervation is required for adult SLG tissue homeostasis by preferentially maintaining and replacing functional acini through regulation of SOX2 and SOX2$^+$ cells.

To determine whether resupplying salivary gland with nerves could rescue acini and SOX2, we examined murine salivary gland 30 days after denervation. Due to the plasticity of the peripheral nervous system, murine salivary glands become reinnervated over time, as shown by the reappearance of TUBB3$^+$ nerves 30 days after transection (Appendix Fig S2A) and re-expression of the neuronal genes Tubb3, Vip, and Vacht (Fig 3B, blue bars; Yawo, 1987). Surprisingly, we found elevated expression of these neuronal genes at day 30 (Fig 3B), suggestive of hyperinnervation, in response to the original injury. Remarkably, upon reinnervation the levels of Sox2 and Aqp5 transcripts and SOX2 protein as well as numbers of SOX2$^+$ and AQP5$^+$ cells and acinar cell size returned to at or above control levels (Fig 3B and Appendix Fig S2A–C).

To ensure that SOX2$^+$ cells of the SLG were capable of responding directly to acetylcholine produced by the parasympathetic nerves, we analyzed expression of acetylcholine/muscarinic receptors by SOX2$^+$ cells as well as their ability to respond to muscarinic agonists in vivo. SOX2$^+$ cells expressed both CHRM1 and CHRM3 (95 and 99%, respectively; Fig 4A and B) and short-term treatment of wild-type mice with the muscarinic agonist pilocarpine (delivered I.P. and sacrificed at 18 h) increased the percentage of proliferating SOX2$^+$ cells (SOX2$^+$Ki67$^+$ cells; Fig 4C and D). The percentage of SOX2$^+$ cells, however, was not significantly changed by 18 h post-injection (Fig 4D), indicating that muscarinic activation does not induce ectopic expression of SOX2. Thus, these data support our hypothesis that parasympathetic nerves via acetylcholine muscarinic signaling maintain SOX2$^+$ progenitors and acini and promote acinar cell replenishment.

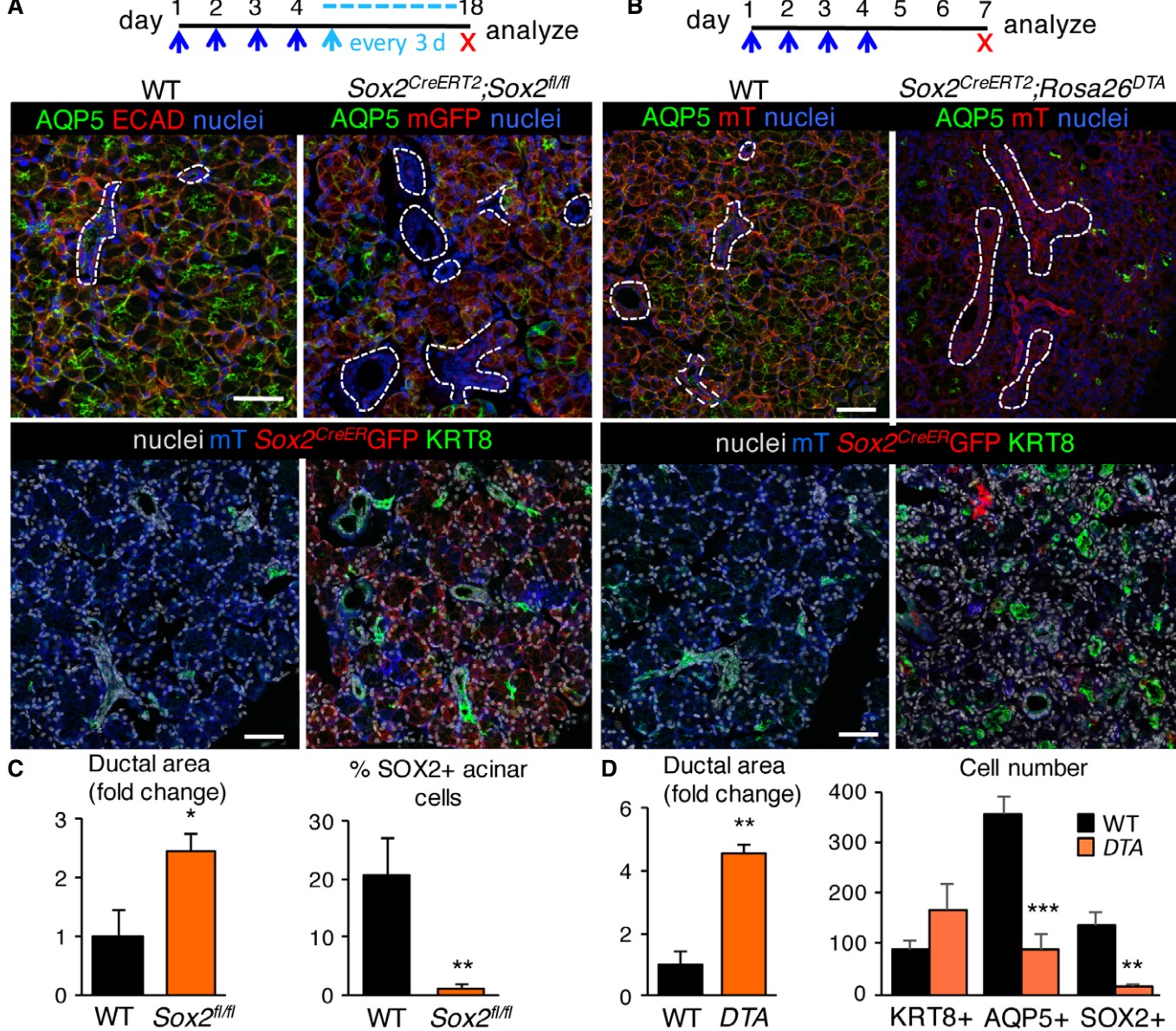

**Figure 2. SOX2 and SOX2+ cells are essential for the replenishment of salivary acinar cells.**

A, B  *Sox2* or SOX2+ cells were ablated in SLG of *Sox2^{CreERT2};Sox2^{fl/fl};Rosa26^{mTmG/+}* mice (A; see schematic) or *Sox2^{CreERT2}Rosa26^{DTA};Rosa26^{mTmG/+}* mice (B; see schematic). Sections were immunostained for AQP5, KRT8, or ECAD and nuclei. Scale bars = 50 μm. Dashed white lines outline ducts. *n* = 3 per genotype.

C, D  Quantification of ductal area in *Sox2^{CreERT2};Sox2^{fl/fl}* (C) or *Sox2^{CreERT2}Rosa26^{DTA};Rosa26^{mTmG/+}* SLG (D) expressed as a percentage of total epithelial area. In (C), right graph, the number of SOX2+ cells in *Sox2^{CreERT2};Sox2^{fl/fl}* SLG expressed as a percentage of total cells. In (D), right graph, the total number of KRT8+ ductal, AQP5+ and SOX2+ acinar cells in wild-type and *Sox2^{CreERT2}Rosa26^{DTA};Rosa26^{mTmG/+}* mice was counted.

Data information: Calculations of cell numbers/duct areas were performed on three non-consecutive fluorescent sections of each SLG from *n* = 3 mice/genotype. Data in (C and D) (*n* = 3) are means + SD and were analyzed by Student's *t*-test. In (C), \**P* = 0.011 and \*\**P* = 0.0041, and in (D), left graph \*\**P* = 0.0015 and right graph \*\*\**P* = 0.0007 and \*\**P* = 0.0018.

**Murine salivary glands regenerate after radiation-induced damage via SOX2**

The murine salivary gland (predominantly the SMG) has been used extensively to investigate the effects of ionizing radiation (IR) on glandular function and structure, where typical analysis is limited to degenerative responses. The C57BL/6 background has

been reported to undergo loss of acinar cells and a decline in salivary flow rate after a single 10 Gy dose (Zeilstra *et al*, 2000; Coppes *et al*, 2001, 2002). However, the regenerative capacity of the tissue and whether it remains innervated after IR are unknown. To test the effect of radiation on SOX2-mediated salivary gland regeneration, we analyzed innervation, SOX2+ cells and SOX2-mediated acinar cell replenishment in murine SLG after

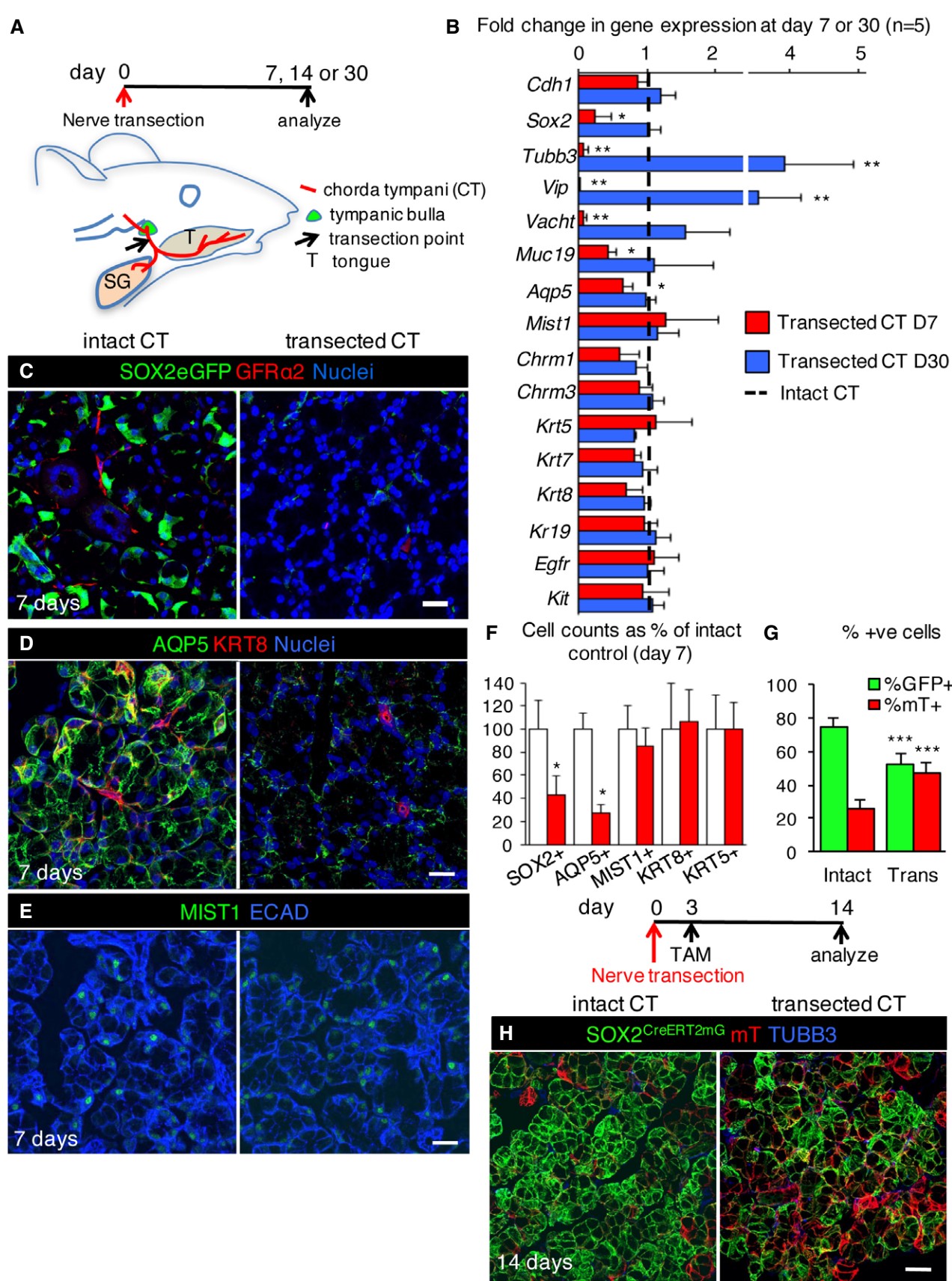

**Figure 3.**

**Figure 3.  Parasympathetic nerves are necessary for maintaining SOX2+ cells and promoting SOX2-mediated acinar cell replacement.**
A       Schematic shows time course of denervation and location of chorda tympani (CT) in adult mice.
B       Gene expression (qPCR) analysis of intact (uninjured contralateral gland) and nerve transected SLG 7 and 30 days (D7 or D30) after surgery. Gene expression was normalized to *Rsp18* and intact controls for each time point.
C–F     Control and nerve transected SLG were immunostained 7 days after denervation for nerves (GFRα2), acinar cells (AQP5 and MIST1), ductal cells (KRT8), and epithelial cells (ECAD). The number of SOX2+, AQP5+, MIST1+, KRT8+, and KRT5+ cells in control and transected SLG was counted and represented as a percentage of the number of cells in control SLG (F). Scale bars in (C, D and E) = 25 μm.
G, H    Recombination was induced in *Sox2CreERT2;Rosa26mTmG* mice 3 days after nerve transection and SLG traced for 11 days before being immunostained for TUBB3. The percentage of GFP+ and mT+ acinar cells in control and transected glands are shown in (G). Scale bar in (H) = 25 μm.

Data information: Data in (B) (*n* = 5) are means + SEM and were analyzed using a one-way analysis of variance with a *post hoc* Dunnett's test. *Sox2* (D7) *P = 0.0455, *Tubb3* (D7) **P = 0.0082, *Tubb3* (D30) **P = 0.0091, *Vip* (D7) **P = 0.0098, *Vip* (D30) **P = 0.0063, *Vacht* (D7) **P = 0.0071, *Muc19* (D7) *P = 0.0419, *Aqp5* (D7) *P = 0.0468. Data in (F and G) were calculated from three non-consecutive fluorescent sections of each SLG from *n* = 5 mice/group or genotype, are means + SD, and were analyzed by Student's *t*-test. SOX2+ *P = 0.0197, AQP5+ *P = 0.0106, %GFP+ ***P = 0.0000096, %mT+ ***P = 0.0000096.

a single dose of gamma-radiation to the head and neck. Similar to previous studies in salivary glands (Avila *et al*, 2009), we found a 10 Gy dose induces DNA damage and cell cycle arrest, as well as reduces cell proliferation in the SLG in the first day following IR, as shown by a substantial increase in the pro-apoptotic gene *Bax* and the cell cycle inhibitor *Cdkn1a* (p21) (Fig EV4A) and a reduction in transcript levels of the cell proliferation marker *Mki67* (Fig EV4A). We then measured changes in innervation and nerve function in IR SLG by immunolabeling for TUBB3+ nerves and performing semi-quantitative PCR (qPCR) for *Tubb3* and the parasympathetic nerve-derived neurotransmitter *Vip*. As shown in Fig 5A and B TUBB3+ nerves were unchanged in the IR SLG compared to non-IR controls 1 and 3 days after radiation. However, transcript levels of *Tubb3* and *Vip* were significantly reduced at 1 and 3 days following IR, suggesting nerve function is reduced at early stages (Fig EV4B). Similarly, transcripts for *Sox2*, *Mist1,* and *Aqp5* were reduced immediately following IR but returned to control levels by day 7 post-IR (Fig EV4B). The number of SOX2+ cells was significantly reduced at day 1 post-IR (Fig 5C), while the number of CCND1+SOX2+ cells was significantly increased at days 3 and 7 post-IR (Fig 5C). In order to determine whether SOX2-mediated cell replacement was affected following IR-induced damage, we analyzed the extent of SOX2-mediated replenishment in *Sox2CreERT2;Rosa26mTmG* mice 14 days post-IR. Strikingly, we found acinar cells were replaced by SOX2+ cells in the IR SLG (GFP+ cells) similar to the non-IR control with the production of both SOX2+GFP+ and SOX2-negative progeny (Fig 5D, white arrowheads indicate SOX2-negative progeny), suggestive of the presence of a transit-amplifying cell which is no longer SOX2+ but derived from a SOX2+ cell (i.e., lineage-traced). Furthermore, the number of SOX2+ cells in the IR SLG was similar to controls by day 14 (Fig 5D). Thus, despite an initial loss of nerve signaling, SOX2, and acinar cell markers, the acinar compartment is capable of being replenished by SOX2+ cells after radiation-induced damage.

Next, we tested whether the SLG could regenerate following IR injury in the absence of *Sox2*. As shown in Fig 5E, *Sox2CreERT2;Sox2fl/fl* mice that had been irradiated with a single 10 Gy dose were unable to repopulate the tissue with functional AQP5+ acini. Indeed, in the absence of *Sox2*, we observed a loss of AQP5+ acinar cells and disrupted tissue architecture compared to wild-type mice at 14 days post-IR (Fig 5E). This outcome further confirms that *Sox2* is essential for SLG regeneration following radiation-induced injury.

## SOX2+ cells can replenish the irradiated salivary gland in response to cholinergic mimetics

As our data suggest cholinergic cues replenish the acinar lineage in the SLG, we determined whether SOX2+ cells can replenish acini in response to muscarinic activation in healthy and irradiated SLG using our *ex vivo* lineage tracing model. As shown in Fig 6A and B, there was an increase in GFP+ clones in healthy SLG tissue cultured with the acetylcholine mimetic carbachol (CCh) for 48 h. This increase in GFP+ clones was associated with an increase in cell proliferation (Ki67+ cells) with CCh treatment (Fig EV4E). In our IR model, recombination was induced in *Sox2CreERT2; Rosa26mTmG* mice 24 h before animals were subjected to a single dose of IR (Fig 6A). This time point was chosen as a lag time of 12–24 h has been previously reported for tamoxifen-induced recombination of Cre lines in mice (Nakamura *et al*, 2006). Thus, single SOX2+ cells are labeled by 24 h after injection (Fig 6B, see 0 h panels). SLGs were collected within 1 h of radiation exposure and cultured *ex vivo* with or without CCh for 48 h. As shown in Fig 6B and quantified in Fig 6C, GFP+ clones were more abundant in IR SLG compared to the no IR controls, suggesting IR activated SOX2+ cells to repopulate the tissue. This was likely due to cholinergic signaling from remaining nerves as IR explants cultured with the muscarinic receptor antagonist 4-DAMP exhibited similar acinar cell replenishment to non-IR cultured salivary gland (Fig 6B and C). Importantly, treatment of IR explants with CCh increased GFP+ cells compared to IR alone (Figs 6B and C, and EV4C and D). Based on these data, we conclude that SOX2+ cells are capable of repopulating the IR SLG in response to muscarinic activation.

## Acetylcholine/muscarinic signaling maintains SOX2 and the acinar lineage in human SG

For humans, IR irreversibly reduces parasympathetic (and increases sympathetic) innervation in SG (Knox *et al*, 2013; Fig EV5A; *GFRA2*), as well as markers of the acinar lineage (*AQP3, MIST1, AMY1*) but not ductal lineage (*EGFR, KRT19*) compared to no IR controls (Fig 7A; IR was delivered ~2 years prior to surgery, *n* = 7 IR and 11 no IR). Lineage markers were confirmed in human tissue (SMG was used due to availability) by immunofluorescence (Fig EV5B). Furthermore, we found transcript levels for *SOX2*, *GFRA2, CHRM1,* and *CHRM3,* but not tyrosine hydroxylase (*TH,* sympathetic marker), were also significantly downregulated or trended toward downregulation following IR (Figs 7A and EV5A),

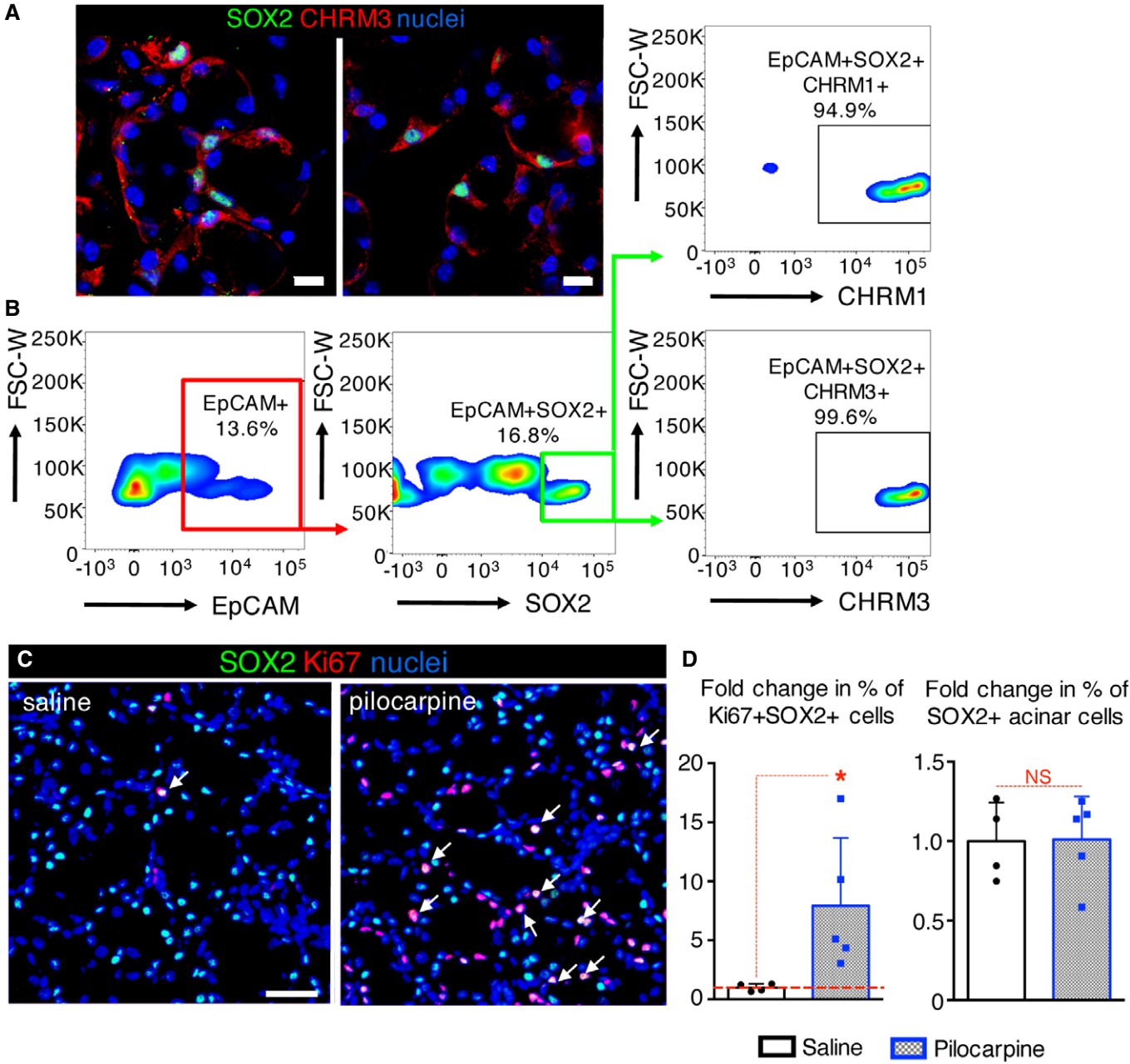

**Figure 4. Muscarinic signaling promotes SOX2$^+$ cell proliferation.**

A   Adult SLG was immunostained for SOX2, CHRM3, and nuclei. Images are a 6 μm (left) and 1 μm (right) projection of 1-μm and 0.175-μm confocal sections. Scale bar = 10 μm.

B   The percentage of epithelial SOX2$^+$ cells that are CHRM1$^+$ or CHRM3$^+$ were counted using flow cytometry and expressed as a percentage of total EpCAM$^+$SOX2$^+$ cells.

C, D   Adult SLG from mice treated with pilocarpine or saline (control) was immunostained for SOX2, Ki67, and nuclei. White arrows indicate proliferating SOX2$^+$ (SOX2$^+$Ki67$^+$) cells (C). Scale bar = 20 μm. (D) The fold changes in % of SOX2$^+$ and SOX2$^+$Ki67$^+$ cells with pilocarpine treatment.

Data information: (B) SLG were pooled from $n$ = 3 mice (10,000 events). Data in (D) were calculated from three non-consecutive fluorescent sections of each SLG from $n$ = 4 (saline) or $n$ = 5 (pilocarpine) mice, are means + SD, and were analyzed by Student's $t$-test. *$P$ = 0.0487.

indicating parasympathetic function as well as the ability of cells to respond to acetylcholine was depleted. Thus, we hypothesized that the loss of regenerative capacity of human salivary glands following IR is due to reduced parasympathetic innervation of the SOX2$^+$

progenitor cells, and that acetylcholine/muscarinic signaling is sufficient to maintain *SOX2* expression and promote the acinar lineage. To test this hypothesis, we established a novel human explant-murine nerve co-culture system. In this model, non-irradiated

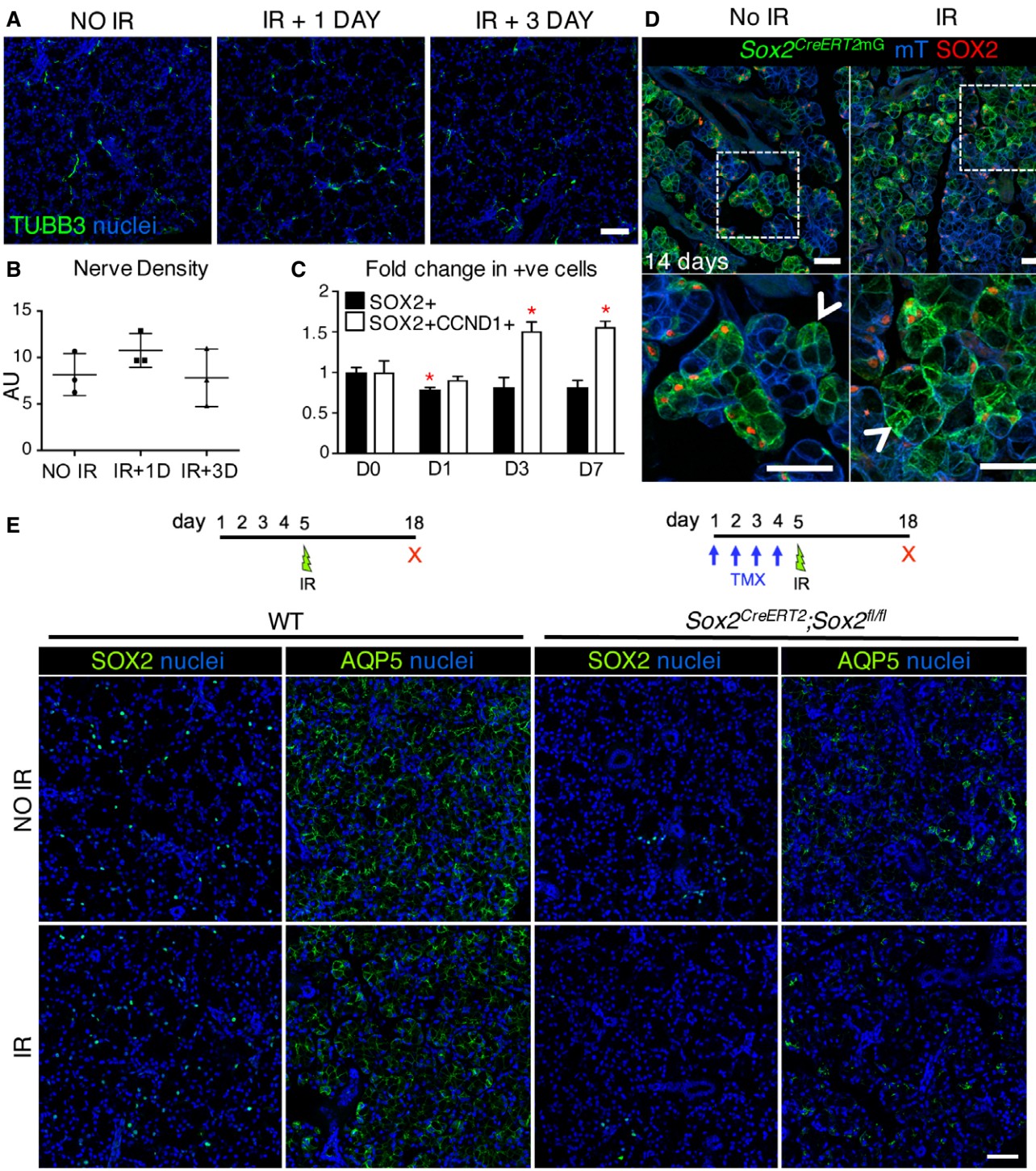

**Figure 5.  *Sox2* is essential for SLG regeneration following radiation injury.**

A–D    Representative images of control (0 Gy; no IR) and irradiated (10 Gy; IR) SLG from wild-type (A) and *Sox2^CreERT2^;Rosa26^mTmG^* (D) mice and analyzed 1, 3, and 14 days later. (A–C) SLG was stained for nerves (TUBB3), SOX2, CyclinD1 (CCND1), and nuclei (A, C), and nerve density calculated (B). (C) The number of SOX2⁺ and CCND1⁺ cells was quantified. (D) *Sox2^CreERT2^;Rosa26^mTmG^* mice were traced for 14 days post-irradiation (IR) and immunostained for SOX2 (D, red). White arrowheads indicate SOX2-negative progeny. Scale bars in (A and D) are 50 μm.

E    *Sox2^CreERT2^;Sox2^fl/fl^* mice and wild-type littermates were irradiated with 10 Gy IR and SLG analyzed 13 days later. SLG was immunostained for SOX2, AQP5, and nuclei. Scale bar = 50 μm.

Data information: Data in (B) are means + SD, *n* = 3 with individual values plotted. Data in (C) were calculated from three non-consecutive fluorescent sections of each SLG from *n* = 3 mice/treatment, are means + SD, and data were analyzed using a one-way analysis of variance with *post hoc* Dunnett's test. SOX2⁺ (D1) *\*P* = 0.0487, SOX2⁺CCND1⁺ (D3) *\*P* = 0.318, SOX2⁺CCND1⁺ (D7) *\*P* = 0.0291.

human SMG is dissected into < 1-mm pieces and placed alongside an embryonic day 13 murine submandibular gland parasympathetic ganglia (contains mesenchyme) or mesenchyme only (control, no nerves). Tissues are then co-cultured on a filter floating above serum-free media for 7 days. Nerves migrated in and around the tissue (Fig 7B) and actively maintained tissue structure as shown by higher levels of CDH1 (E-cadherin, gene, and protein expression) in the explants co-cultured with nerves compared to mesenchyme alone (Fig 7C and D). We analyzed the explants for the neuroattractants neurturin (*NRTN*) and nerve growth factor (*NGF*) (Knox *et al*, 2013) in an attempt to ascertain what factors induced this nerve migration. However, we found no increase in expression in the

presence of the ganglia compared to the mesenchyme alone (Fig 7D), suggesting that either other neuroattractants were being synthesized, that the epithelia produces these neuroattractants whether the nerves are present or not (i.e., the mesenchyme is supportive enough to maintain epithelial homeostasis) or that this phenomenon occurs earlier in culture (i.e., at an earlier stage when the nerves are just starting to envelop the epithelia) and any differences have been resolved by this later time point (day 7). Strikingly, glandular explants cultured with nerves also exhibited increased transcript levels of *SOX2* (~1.8-fold to twofold) and the acinar markers *MIST1* and *CD44*, muscarinic receptor *CHRM3* (also expressed by acinar cells; Giraldo *et al*, 1988) but less consistently

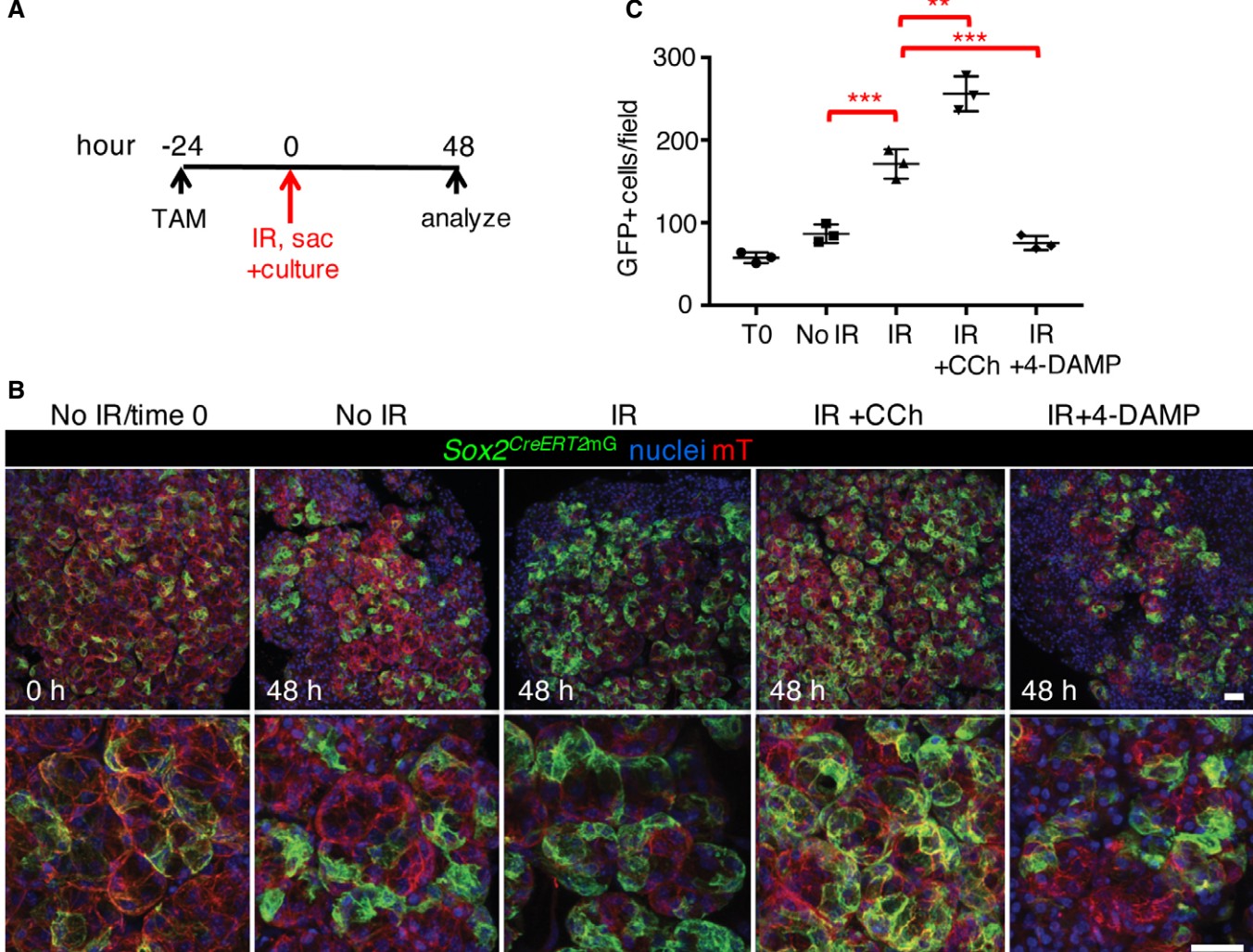

**Figure 6. SOX2⁺ progenitors can replenish acinar cells after radiation-induced damage in response to muscarinic stimulation.**

A  SLG from *Sox2^{CreERT2}*;*Rosa26^{mTmG}* mice was collected following a single 15 Gy dose of IR and explants cultured for 0–48 h. Schematic shows the timing of recombination, culture, and analysis.
B  Representative images of lineage-traced explants immunostained for nuclei. Scale bar is 50 μm.
C  Quantification of the number of GFP⁺ cells.

Data information: Data in (C) were calculated from three random areas of *n* = 3 immunostained explants for each treatment with individual values plotted, and data were analyzed using a one-way analysis of variance with *post hoc* Dunnett's test. Error bars show mean ± SD. No IR vs. IR ***P = 0.0022, IR vs. IR + CCh **P = 0.0058, IR vs. IR + 4-DAMP ***P = 0.0010.

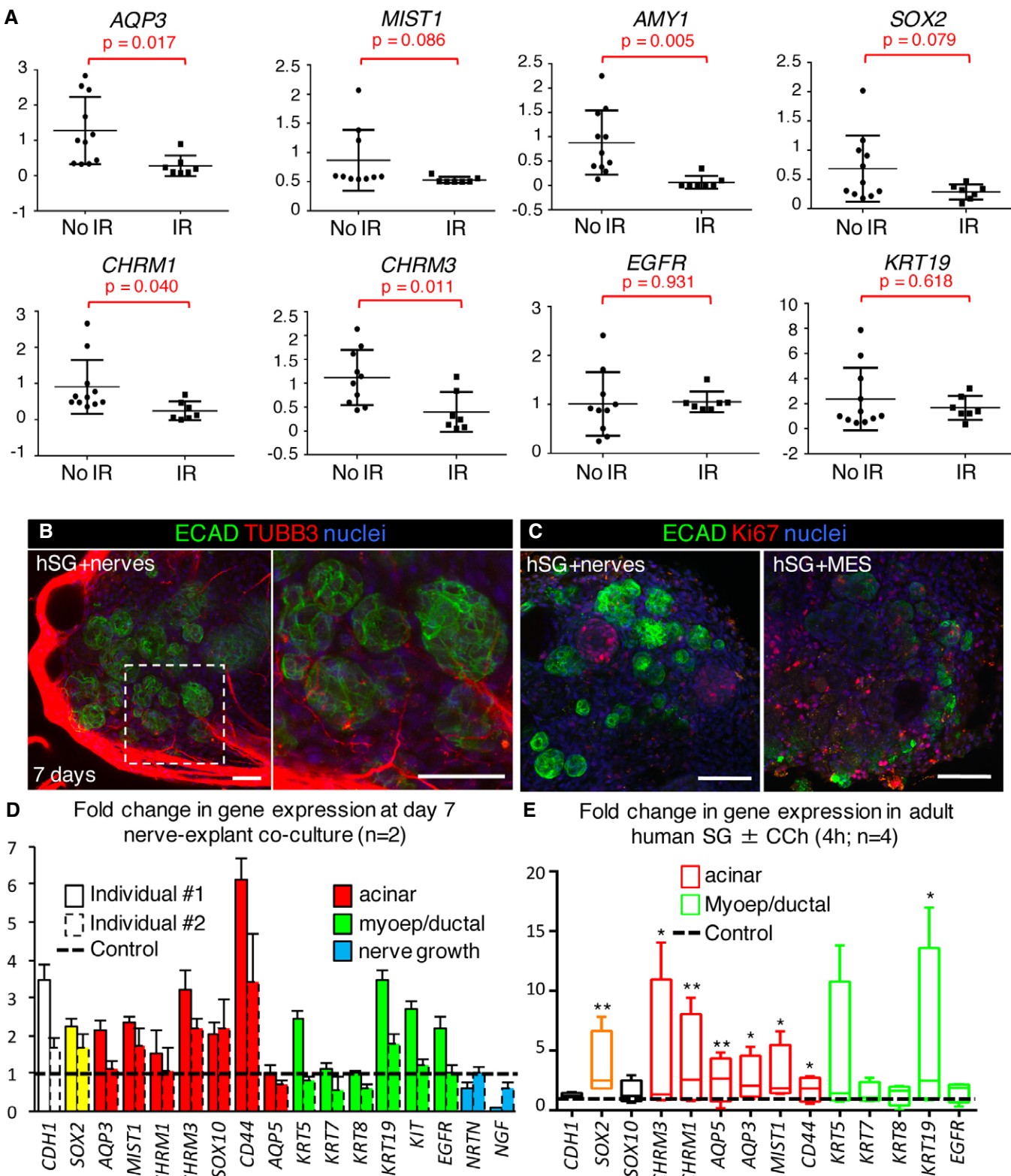

**Figure 7.**

the ductal genes *KRT19*, *KIT*, and *EGFR* compared to the mesenchyme only controls (Fig 7D, data normalized to mesenchyme only, *n* = 2 separate individuals). In addition,

epithelial cell proliferation, marked by the presence of Ki67+ cells, was increased in the presence of the nerves (Figs 7C and EV5C). Together, these data suggest that parasympathetic nerves are able to

**Figure 7.  Acetylcholine/muscarinic signaling maintains SOX2 and the acinar lineage in human SG.**

A   Human salivary gland obtained from healthy individuals (no IR; submandibular) or patients who received radiation therapy for head and neck cancer (IR) were subjected to qPCR.

B–D   Human SMG explants cultured for 7 days with either murine embryonic day (E) 13 parasympathetic ganglia (nerves) or E13 mesenchyme (MES). Explants were analyzed by immunostaining for markers of nerves (B, TUBB3) or cell proliferation (C, Ki67) or by qPCR (D). Scale bars = 50 μm (B, C).

E   qPCR analysis of adult human salivary gland (SMG or PG) explants of four different individuals cultured for 4 h ± CCh (200 nM, n = 4). Individual datasets are shown in Fig EV5D.

Data information: Data in (A) n = 11 for no IR and n = 7 for IR; 30–85 years. Data were normalized to *GAPDH* with individual values plotted and analyzed using a Student's *t*-test with a false discovery rate set to 0.05. Error bars show mean ± SD. *AQP3* P = 0.017, *MIST1* P = 0.086, *AMY1* P = 0.005, *SOX2* P = 0.079, *CHRM1* P = 0.040, *CHRM3* P = 0.011, *EGFR* P = 0.931, *KRT19* P = 0.618. Data in (D) are means + SD of n = 2 individuals where solid line columns represent individual #1 and broken line columns represent individual #2. Data were normalized to salivary glands from the same individuals cultured with murine mesenchyme (Control, black dashed line). Data in (E) are box and whisker plots of n = 4 different individuals, showing means (horizontal line), upper and lower quartiles (box) and upper and lower values (whiskers). Data were normalized to the untreated control (black dashed line). n = 5–8 explants per individual. Data were analyzed using a one-way analysis of variance with *post hoc* Dunnett's test. *SOX2* \*\*P = 0.00834, *CHRM3* \*P = 0.0449, *CHRM1* \*\*P = 0.0093, *AQP5* \*\*P = 0.0069, *AQP3* \*P = 0.0375, *MIST1* \*P = 0.0379, *CD44* \*P = 0.0485, *KRT19* \*P = 0.0461.

maintain SOX2 expression and acinar and ductal markers in the human salivary gland.

To address whether acetylcholine/muscarinic signaling is sufficient to maintain acini and SOX2$^+$ cells in human salivary gland, we subjected non-irradiated human SMG or PG (both express SOX2) tissue explants to muscarinic stimulation in an *ex vivo* system. Culture of patient-derived human tissue from four separate individuals (n = 4) with CCh significantly increased expression of *SOX2*, muscarinic receptors *CHRM1* and *CHRM3* [both expressed by adult acinar cells (Giraldo *et al*, 1988; Mei *et al*, 1990)], as well as *AQP3*, *AQP5* (Gresz *et al*, 2001), and *MIST1* within 4 h of muscarinic stimulation (Fig 7E, n = 4, individual datasets shown in Fig EV5D). While surgical denervation does not adversely affect expression of *Mist1* (Fig 3B and E), muscarinic stimulation is sufficient to increase *MIST1* in human cultures and suggests that although not required for acinar cell identity, acetylcholine/muscarinic signaling may act as a positive regulator of the secretory program. The variability in response between the four patient-derived samples is likely due to biological diversity between human patients, differences in the type of gland sourced (SMG and PG used), and the age of the patient (age of donor ranges from 30 to 78 years). However, in all cases, we observed an increase in *SOX2* and a number of acinar markers in the presence of CCh. Although we also measured an increase in *KRT19* gene expression with CCh, the other exclusive ductal markers *EGFR*, *KRT7*, and *KRT8* remained unchanged, suggesting ductal cells do not generally respond to muscarinic agonists (Fig 7E). Given nerves increased ductal genes in the human salivary gland in co-culture, other factors may be produced by nerves to elicit changes in the ducts (compare Fig 7D and E). Thus, neuronal acetylcholine/muscarinic signaling is sufficient to promote the acinar lineage and maintain SOX2 expression in adult human salivary gland.

## Discussion

Our study reveals SOX2$^+$ cells as progenitors in the adult salivary gland essential to the replenishment of acini with the unexpected capacity to repopulate the tissue after radiation-induced damage. We further show that cholinergic nerves play a vital role in controlling SOX2-mediated acinar cell replacement during homeostasis and that this neuronal influence can be replicated through addition of cholinergic mimetics to irradiated tissue. Thus, contrary to current dogma that murine salivary glands do not regenerate after radiation-induced damage (Zeilstra *et al*, 2000; Coppes *et al*, 2001, 2002), these data indicate that, at least in mice, salivary glands have extensive regenerative capacity after radiation-induced damage. Furthermore, as we find that the acinar lineage (and SOX2) in human tissue is also responsive to cholinergic mimetics, targeting SOX2$^+$ cells and maintaining cholinergic nerves may aid in the recovery of functional salivary acini after damage by radiation therapy.

SOX2 has an essential role in organism development and regulates the homeostasis of epithelial tissues such as the stomach, trachea, and intestine (Arnold *et al*, 2011). Similar to the reduced homeostatic capacity of adult tracheal cells after *Sox2* ablation (Que *et al*, 2009), we found depletion of AQP5$^+$ acinar cells after ablation of *Sox2* in SOX2$^+$ acinar progenitors under both homeostatic and damage conditions. Intriguingly, although atrophied, we did not find increased cell death in the absence of *Sox2* but substantially fewer cycling acinar cells, suggesting that upon ablation, these cells exit the cell cycle and undergo differentiation to produce dormant AQP5-deficient acinar cells. The striking loss of acini upon ablation of SOX2$^+$ cells further confirmed that these are the sole progenitors for the acinar lineage in the SLG and is consistent with a recent report demonstrating pancreas acinar cells are not equipotent but contain a subset of progenitors (Wollny *et al*, 2016). Our results contrast to a recent report by Aure and co-workers showing that salivary acinar cells replenish through self-duplication of mature cells rather than via progenitor differentiation (Aure *et al*, 2015), similar to pancreatic beta cells (Dor *et al*, 2004). The conclusion that tissue repopulation is due to self-duplication of salivary acinar cells was based on the use of an inducible *Mist1* promoter that is expressed by all acinar cells, including SOX2$^+$MUC19$^-$ cells. As such, subsets of cells traced upon recombination likely include progenitors; however, no analysis was conducted to determine whether these subsets possess a stem cell signature.

As with all epithelial organs, peripheral nerves are required to maintain the structural homeostasis of rodent and human salivary gland (Schneyer & Hall, 1967; Mandour *et al*, 1977; Wang *et al*, 1991; Fu & Gordon, 1995; Lujan *et al*, 1998; Kang *et al*, 2010; Batt & Bain, 2013). During organogenesis, parasympathetic nerves maintain a progenitor population that can contribute to the tissue after radiation damage via acetylcholine/muscarinic signaling (Knox *et al*, 2013). Although previous denervation studies in the salivary gland indicated that the organ and acinar cells decrease in size (Schneyer & Hall, 1967; Mandour *et al*, 1977; Kang *et al*, 2010),

whether innervation was required for acinar cell replacement was not known. Here we reveal that nerves directly regulate SOX2 to drive acinar cell replacement from lineage-restricted progenitors. This outcome is consistent with the known role of SOX2 in the regulation of self-renewal and cell fate in many other organs (Arnold *et al*, 2011). However, to date, only cell-intrinsic signaling pathways including those mediated by the WNT, FGF, and EGFR families have been shown to regulate SOX2 (Hashimoto *et al*, 2012; Dogan *et al*, 2014; Rothenberg *et al*, 2015; Lee *et al*, 2016). This extrinsic nerve-based model has the distinct advantage over cell-intrinsic signals in that, unlike their target organs, neurons themselves are highly resistant to radiation damage (Tofilon & Fike, 2000; Wong & Van der Kogel, 2004). Whether this mechanism also regulates the maintenance of other SOX2-expressing epithelial organs, for example, taste buds (Suzuki, 2008), prostate and seminal vesicles (Wanigasekara *et al*, 2004), stomach (Tatsuta *et al*, 1985; Zhao *et al*, 2014), and cornea (Ueno *et al*, 2012), remains to be tested. However, these results suggest cholinergic nerves may function in the regeneration of these tissues.

Previous studies have utilized the murine salivary gland as a model of radiation-induced degeneration (Zeilstra *et al*, 2000; Coppes *et al*, 2001, 2002). These investigations have been based on the assumption that regeneration is impaired after moderate to high doses of radiation, a hypothesis supported by the reduced saliva flow measured in animals receiving radiation (Redman, 2008). However, to date, there has been no *in vivo* analysis of cell replacement after radiation. Our data indicate that murine acinar cells are highly regenerative, at least in the first 30 days after radiation exposure, and are capable of repopulating the acini similar to uninjured controls. It is clear, however, that this regenerative capacity cannot be sustained for the long term as degeneration/senescence in murine salivary glands occurs 3–6 months after radiation (Urek *et al*, 2005; Marmary *et al*, 2016). As such, it is likely that the regenerative capacity of SOX2$^+$ cells does fail eventually and further analysis is required to discern the cause. It also remains to be determined whether the human salivary gland can regenerate in the days/months after therapeutic radiation and if this regenerative capacity fails in the long term due to the absence of SOX2$^+$ cells in combination with parasympathetic nerves. Indeed, a time course analyzing changes in salivary glands from patients is required to understand how these organs are affected in the short term and long term. However, our results suggest that targeting these stem cells and their innervating nerves to control and sustain tissue regeneration in response to radiation damage may provide a means of maintaining/repairing tissue for the long term.

A number of recent studies have aimed to address gland regeneration after radiation damage in the mouse model by either isolating putative stem cell populations for reimplantation (Nanduri *et al*, 2013, 2014) or sparing the regions within the gland that are thought to harbor the stem cells (van Luijk *et al*, 2015). However, the identity of these endogenous stem cells and whether they contribute to the acinar cell compartment was unclear. Furthermore, the effect of such manipulations on salivary gland innervation has not been reported. Based on our study, it is possible that unintentional increases in innervation due to tissue perturbation lead to maintenance and expansion of SOX2$^+$ progenitor cells that regenerate acini to restore salivary function. Xiao *et al* (2014) reported recovery of murine salivary function and architecture after radiation-induced

damage with the addition of glial-derived nerve factor (GDNF), a neuroattractive factor (Knox *et al*, 2013). However, a recent study demonstrated that GDNF itself does not protect SG stem cells from radiation-induced damage directly (Peng *et al*, 2017), suggesting that such outcomes could be the result of improvements to the supporting niche.

Together, our study highlights the extensive regenerative capacity of salivary glands that occurs through the expansion and differentiation of a progenitor cell population, even in the face of genotoxic shock. Based on these data, we propose that by directly targeting SOX2$^+$ cells within the tissue, or by isolating and expanding these cells for transplantation and activation, we might regenerate the secretory units of salivary glands and return quality of life to the patient. This would also require maintenance of parasympathetic nerves, as proposed previously (Knox *et al*, 2013), to sustain the SOX2$^+$ population. Given organs, such as the intestine, glandular stomach, trachea, and taste buds express SOX2, are heavily innervated by the autonomic nervous system and are damaged by therapeutic radiation for the elimination of cancers, such a strategy may be applicable to the repair of multiple organ systems.

# Materials and Methods

### Mouse lines

All procedures were approved by the UCSF Institutional Animal Care and Use Committee (IACUC) and were adherent to the NIH Guide for the Care and Use of Laboratory Animals. Mouse alleles used in this study were provided by The Jackson Laboratory and include *Sox2$^{eGFP}$* (Arnold *et al*, 2011), *Sox2$^{CreERT2}$* (Smith *et al*, 2009), *Sox2$^{fl/fl}$* (Taranova *et al*, 2006), *Rosa26$^{mTmG}$* (Muzumdar *et al*, 2007), *Rosa26$^{DTA}$* (Wu *et al*, 2006), and *Kit$^{CreERT2}$* (Klein *et al*, 2013).

### Animal experiments

Adult female mice (aged between 6 and 8 weeks) were used in all experiments, unless otherwise stated. Mice were housed in the University of California San Francisco Parnassus campus Laboratory Animal Resource Center (LARC), which is AAALAC accredited. Mice were housed in groups of up to five per cage where possible, in individually ventilated cages (IVCs), with fresh water, regular cleaning, and environmental enrichment. Appropriate sample size was calculated using power calculations. For transgenic studies, sample size was restricted by length of time required to breed enough animals of the required genotype and gender. Wild-type animals were randomized into experimental groups using Microsoft Excel software. Transgenic animals were assigned to groups based on genotype. All animals were given a unique ID number and as such were blinded to the researcher during analysis.

### *Genetic ablation of Sox2 or SOX2$^+$ cells*
Conditional ablation of *Sox2* was achieved by injecting *Sox2$^{CreERT2}$; Sox2$^{fl/fl}$:R26$^{mTmG}$* mice with 2.5 mg/20 g tamoxifen each day for four consecutive days, and every other third day, before euthanizing on day 18. Ablation of SOX2$^+$ cells was performed by injecting *Sox2$^{CreERT2}$; Rosa26$^{DTA}$:R26$^{mTmG}$* mice with 2.5 mg/20 g tamoxifen

each day for four consecutive days, before euthanizing on day 7. The $Rosa26^{mTmG}$ mouse provides a valuable tool to lineage trace in conjunction with cell ablation/recombination (Muzumdar *et al*, 2007). In brief, this model consists of a double-fluorescent Cre reporter that expresses membrane-targeted tandem dimer Tomato (mT) prior to Cre-mediated excision and membrane-targeted green fluorescent protein (mG) after excision (Fig EV1D). Thus, in these mice, cells where recombination has occurred and are thus lacking *Sox2* or expressing DTA will express GFP. Endogenous GFP was imaged in experiments where cryosections were used, while a GFP antibody was used for paraffin-embedded tissue (chicken anti-GFP; 1:500, Aves Labs, GFP-1020).

### Lineage tracing of SOX2$^+$ cells

$Sox2^{CreERT2};Rosa26^{mTmG}$ mice were injected with 2.5 mg tamoxifen and euthanized after 24 h, 14 days, or 30 days.

### Lineage tracing of KIT$^+$ cells

$Kit^{CreERT2};Rosa26^{mTmG}$ mice were injected with 2.5 mg tamoxifen daily for four consecutive days and euthanized after 14 days or 6 months.

### In vivo denervation experiments

C57BL/6 or *Sox2eGFP* mice were administered with analgesics 30 min prior to surgery (carprofen and buprenorphine; Patterson Veterinary and Buprenex; 0.1 and 100 mg/kg (IP), respectively) and anesthetized via inhalation with 2% isoflurane/$O_2$ mix. The surgical area was shaved and prepared for incision with alternating iodine and alcohol washes, followed by local anesthetic (lidocaine; Hospira Inc., 8 mg/kg). An incision was made anterior to the ear and the chorda tympani located as previously described (Klimaschewski *et al*, 1996) and completely transected using spring scissors. The skin was sutured using non-absorbable silk sutures (Ethicon) and the wound further covered by surgical adhesive (Vetbond). The contralateral nerve was left intact as a control. Mice were euthanized after 7 or 30 days.

### Pilocarpine experiments

Adult male C57BL/6 mice (aged between 6 and 8 weeks) were treated with pilocarpine (Sigma-Aldrich, P0472; 0.68 mg/ml in 0.9% sterile saline). Mice were anesthetized using isoflurane and injected I.P. with pilocarpine at a dose of 4.5 mg/kg bodyweight (200 μl), or 0.9% saline as a vehicle control. Mice were euthanized after 18 h and glands processed for immunofluorescent analysis.

### Gamma-radiation experiments

C57BL/6 mice were anesthetized with 1.25% 2,2,2-tribromoethanol (Alfa Aesar) in 0.9% saline (Vedco Inc.). The mice were irradiated using a $^{137}$Cs source by being placed into a Shepherd Mark-I-68A $^{137}$Cs Irradiator (JL Shepherd & Associates). Two lead blocks, positioned 1.5 cm apart, were used to shield the body and the very anterior part of the mouth (the snout) of the mice and expose only the neck and part of the head. The 1.5 cm opening was centered at position 3 (20 cm from the $^{137}$Cs source, 15.5 cm from the width of the irradiator cavity). Mice were exposed to two doses of 5 Gy at a dose rate of 167 Rads/min for 2.59 min (one of each side of the head,

bilateral, and sequential but on the same day) for a total dose of 10 Gy, to irradiate the salivary glands. This dose was calculated by isodose plot mapping (dose distribution), provided by the manufacturer, and EBT films (Brady *et al*, 2009) were used to localize the 100% region of exposure for mouse placement. Control mice were anesthetized as per experimental mice, but did not undergo radiation treatment. All mice were allowed to completely recover before returning to normal housing and were given soft diet *ad libitum* (ClearH2O). Mice were euthanized after 1 h or 1, 3, 7, 14, or 30 days.

### Organ culture experiments

#### Ex vivo linage tracing in adult salivary gland

Salivary glands from $Sox2^{CreERT2};Rosa26^{mTmG}$ mice (recombination induced 24 h prior) were mechanically dissected into < 1-mm pieces, placed in complete media in the presence or absence of 200 nM CCh (Sigma-Aldrich) or 10 μM 4-DAMP (Tocris), and cultured for 48 h before being fixed for immunofluorescence. In some cases, mice were irradiated with three doses of 5 Gy before glands were taken for culture as above.

#### Human salivary gland tissue isolation and culture

Adult human salivary gland was obtained from discarded, non-identifiable tissue with consent from patients undergoing neck resection (age 28–78, male and female). Informed consent was given by all subjects and experiments conformed to the principles set out in the WMA Declaration of Helsinki and the Department of Health and Human Services Belmont Report. Patients had either had no irradiation therapy (non-IR) or fractionated radiation therapy (IR) < 2 years prior to surgery. Tissue was immediately placed in 4% PFA, RNA*later* (Qiagen) or DMEM (Thermo Fisher) for live cell explant culture. For *ex vivo* culture, non-IR tissue was dissected into < 1-mm pieces and cultured in serum-free DMEM/F12 containing holotransferrin and ascorbic acid. For explant assays, the tissue (SMG and PG) was incubated with 50–200 nM CCh (200 nM CCh shown in results) for 4 h before being lysed for RNA. For salivary gland explant-parasympathetic submandibular ganglia (SMG) co-culture, tissue was dissected into < 1-mm pieces and cultured on a floating filter above serum-free media. E13 mouse parasympathetic submandibular ganglia were isolated as previously described (Knox *et al*, 2010). One parasympathetic submandibular ganglion per explant was placed next to the human salivary gland and cultured for 7 days and either fixed for immunofluorescent analysis or lysed for RNA.

#### Tissue processing

After fixation, salivary glands (human and mouse) were either processed for OCT or paraffin embedding. For generation of frozen sections, tissue was incubated in increasing concentrations of sucrose (25–75%) and embedded in OCT. 12-μm sections were cut using a cryostat (Leica) and stored at −20°C. Tissue for paraffin was dehydrated by incubating in increasing concentrations of ethanol and subsequently Histo-Clear (National Diagnostics) before embedding in paraffin wax (Sigma-Aldrich). 12-μm sections were cut using a microtome (Leica) and stored at room temperature.

## Immunofluorescent analysis

Whole-mount salivary gland and tissue section immunofluorescence analysis has been previously described (Knox *et al*, 2010). In brief, tissue was fixed with either ice-cold acetone/methanol (1:1) for 1 min or 4% PFA for 20–30 min followed by permeabilizing with 0.1–0.3% Triton X. Tissue was blocked overnight at 4°C with 10% Donkey Serum (Jackson Laboratories, ME), 1% BSA (Sigma-Aldrich), and MOM IgG-blocking reagent (Vector Laboratories, CA) in 0.01% PBS–Tween 20. Salivary glands were incubated with primary antibodies overnight at 4°C: goat anti-SOX2 (1:200, Neuromics, GT15098); goat anti-SOX10 (1:500, Santa Cruz Biotechnology, sc-17342); mouse anti-TUBB3 (clone TUJ1 at 1:400, R&D Systems, MAB1195); goat anti-GFRα2 (1:100, R&D Systems, AF429); rabbit anti-tyrosine hydroxylase (1:100, Millipore, AB152); rat anti-E-cadherin (1:300, Life Technologies, 13-1900); rabbit anti-EGFR (1:200, Abcam, ab52894); rabbit anti-KRT5 (1:1,000, Covance, PRB-160P); rat anti-KRT8 (1:200, DSHB, troma I); mouse anti-KRT7 (1:50, Covance, MMS-148S); rat anti-CD44 (1:200, BioLegend, 103001); mouse anti-Ki67 (1:50, BD Biosciences, 550609); rabbit anti-CCND1 (1:200, Abcam, ab16663); rabbit anti-caspase-3 (1:100, Invitrogen, 34-1700); rabbit anti-AQP3 (1:400, Lifespan Biosciences Inc., LS-B8185); rabbit anti-AQP5 (1:100, Millipore, AB3559); goat anti-MUC19 (1:200, Abcore, AC21-2396); mouse anti-αSMA (1:400, Sigma-Aldrich, C6198); chicken anti-GFP (1:500, Aves Labs, GFP-1020); rabbit anti-CHRM3 (1:1,000, Research and Diagnostics, AS-3741S); and rabbit anti-MIST1 (1:500, gift from Stephen Konieczny, Purdue University). Antibodies were detected using Cy2-, Cy3-, or Cy5-conjugated secondary Fab fragment antibodies (Jackson Laboratories) and nuclei stained using Hoechst 33342 (1:1,000, Sigma-Aldrich). Fluorescence was analyzed using a Leica Sp5 confocal microscope and NIH ImageJ software.

## Morphometric analysis and cell counts

For immunofluorescent analyses (e.g., Fig 2D), cells positively stained for markers were counted using ImageJ. Acinar cell size was measured using ImageJ. All data were obtained using 3–5 fields of view/group, and each experiment was repeated three times. For nerve density analysis, immunofluorescence of nerve marker TUBB3 was analyzed and raw integrated density (displayed as arbitrary units, AU) was calculated using ImageJ.

## Quantitative PCR analysis

RNA was isolated from whole tissue using RNAqueous Micro Kit (Ambion). Total RNA samples were DNase-treated (Ambion) prior to cDNA synthesis using SuperScript reagents (Invitrogen, CA). SYBR green qPCR was performed using 5 ng (mouse) or 4–10 ng (human) of cDNA and primers designed using Primer3 and Beacon Designer software or described on PrimerBank (http://pga.mgh.harvard.edu/primerbank/). Primer sequences are listed in Appendix Tables S1 and S2. Melt curves and primer efficiency were determined as previously described (Hoffman *et al*, 2002). Gene expression was normalized to the housekeeping genes *S18* and *S29* (*Rps18* and *Rsp29*) or *Gapdh* for mouse and *GAPDH* for human and to the corresponding experimental control. Reactions were run in triplicate and experiments performed 2–3 times.

## Flow cytometry

Adult mouse sublingual salivary glands (CD1) were dissected and washed in PBS containing gentamicin (Sigma-Aldrich). Cell

**The paper explained**

**Problem**

Salivary gland acinar cells are inadvertently destroyed by radiation treatment for head and neck cancer, resulting in a lifetime of hyposalivation and co-morbidities. A potential regenerative strategy for replacing injured tissue is the reactivation of endogenous stem cells by targeted therapeutics. However, the identity of these cells in the salivary gland, whether they are capable of regenerating the tissue, and the mechanisms by which they are regulated are unknown.

**Results**

In this study, we show that SOX2[+] cells are essential stem cells for the saliva-synthesizing acinar cells in the murine sublingual gland and that they repopulate acini following radiation injury. SOX2-mediated acinar cell replacement is dependent on parasympathetic nerves and that application of a muscarinic mimetic in the absence of nerves is sufficient to drive regeneration. Although SOX2 is limited to the murine sublingual gland, we show that SOX2[+] cells are present in all three major human salivary glands (submandibular, parotid, sublingual) and that expression of SOX2, along with parasympathetic nerves, is diminished in irradiated biopsies. Crucially, the addition of a muscarinic mimetic increases the expression of SOX2 and markers of gland function in human salivary gland biopsies. Together, these results suggest that salivary tissue degeneration in human radiotherapy patients is due to loss of progenitors and cholinergic signaling.

**Impact**

This study addresses an area of significant unmet clinical need. Based on these data, we propose that by directly targeting SOX2[+] cells within the tissue or by isolating and expanding these cells for transplantation and activation, we might regenerate the secretory units of salivary glands and return quality of life to the patients following irradiation therapy. Given organs, such as the intestine, glandular stomach, trachea, esophagus, and taste buds express SOX2, are heavily innervated by the autonomic nervous system and are damaged by therapeutic radiation for the elimination of cancers, such a strategy may be applicable to the repair of multiple organ systems.

isolation and flow cytometry were performed as previously described (Muench *et al*, 2002; Pringle *et al*, 2011). Briefly, a single cell suspension was created by mincing tissue in HBSS + 1% BSA (Sigma-Aldrich) with a scalpel blade and incubating in a HBSS solution containing 50 mM $CaCl_2$ (Sigma-Aldrich), 40 mg/ml hyaluronidase (Sigma-Aldrich) and 23 mg/ml collagenase type II (Sigma-Aldrich) at 37°C for 15–45 min. The enzyme reaction was quenched by the addition of BSA and the solution filtered through a 70 μm strainer (BD Falcon) and centrifuged at 400 *g* for 8 min. The resulting cell pellet was washed with sterile HBSS + 1% BSA, centrifuged and resuspended in blocking buffer (5% serum and 0.01% $NaN_3$, BioLegend). Cell surface staining was achieved by incubating cell suspensions with antibodies against CD326 (EpCAM; Miltenyi, 130-098-113), rabbit anti-CHRM1 (Research and Diagnostics, AS-3701S), rabbit anti-CHRM3 (Research and Diagnostics, AS-3741S) and rabbit anti-AQP5 (Millipore, AB3559). Subsequently, intracellular staining was achieved following fixation and permeabilization using an intracellular staining kit (eBioscience, 00-5523-00) and antibodies against SOX2 (BD Pharmingen, 562195) and Ki67 (BioLegend, 652405). Flow cytometry was performed on a LSRII (BD) using the appropriate single stained controls and data collected using FACSDiva (BD) and

analyzed using FlowJo. 100,000 events were collected for each sample unless otherwise stated.

## Statistical tests

Normal distribution was assessed using the D'Agostino-Pearson omnibus test. Data were analyzed for statistical significance using Student's *t*-test (unpaired, two groups) or one-way ANOVA (multiple groups) with *post hoc* testing performed using Dunnett or Tukey tests (GraphPad Prism or SPSS). For multiple testing, we used a false discovery rate of 0.05. All graphs show the mean + standard deviation (SD) or mean + standard error of the mean (SEM), as indicated in the legends.

Expanded View for this article is available online.

## Author contributions

EE, AJMay, and SMK designed and planned the study; EE, AJMay, LB, NC-P, SN, and AJMattingly performed the experiments; EE, AJMay, LB, and SMK analyzed and interpreted the data; JLC, WRR, and ADT provided tissue samples; EE and SMK prepared the manuscript, with input from AJMay.

## Acknowledgements

The authors would like to acknowledge Drs. Rushika Perera, Licia Selleri, Rolf Zeller, Andrei Goga, and Scott Oaks for their critical reading of the manuscript. We also acknowledge Alvin Wong for his advice. Funding was provided by a CIRM postdoctoral fellowship (EE) and NIDCR 5R21DE025736 (SMK) and 5R01DE024188 (SMK).

## Conflict of interest

The authors declare that they have no conflict of interest.

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
