## [Review Process File · EMBO Molecular Medicine]

Salivary glands regenerate after radiation injury through SOX2-mediated secretory cell replacement

Elaine Emmerson, Alison J. May, Lionel Berthoin, Noel Cruz -Pacheco, Sara Nathan, Aaron J. Mattingly, Jolie L. Chang, William R. Ryan, Aaron D. Tward, and Sarah M. Knox

Review timeline:

Submission date:	14 May 2017
Editorial Decision (The EMBO Journal):	18 May 2017
Transferred:	18 May 2017
Editorial Decision:	09 June 2017
Revision received:	14 November 2017
Editorial Decision:	04 December 2017
Revision received:	14 December 2017
Accepted:	18 December 2017

Editors: Andrea Leibfried, Roberto Buccione, Céline Carret

Transaction Report:

1st Editorial Decision (The EMBO Journal)

18 May 2017

Thank you for submitting your manuscript entitled 'Salivary glands regenerate after radiation injury through SOX2-mediated secretory cell replacement'. I have read it carefully, and I discussed your work with my colleagues.

Your analysis shows that salivary glands can get repopulated when innervated due to Sox2+ cells. Co-culturing experiments suggest that human salivary glands are better preserved upon radiation if treated with acetylcholine/muscarinic receptor agonists.

We appreciate your findings and the elegant in vivo lineage tracing and culturing assays employed. We also realize that your results are of interest to the field, especially in light of the paper by Aure and colleagues. However, we also noted that Sox2+ cells were previously seen in glands, and that innervation being important for gland function is known as is its positive effect on restoring the tissue via acetylcholine. Furthermore, as you note yourself, SOX2 is known for the regulation of self-renewal and cell fate in many organs. We therefore concluded that your manuscript does not present a sufficiently striking conceptual advance for publication in The EMBO Journal. We are thus returning your manuscript at this time so that the work may be considered elsewhere without further delay.

Given the potential clinical relevance of your findings, I also took the liberty to discuss your work with my colleague Dr. Roberto Buccione, editor at our sister journal EMBO Molecular Medicine. Roberto would like to seek external advice on the suitability of your work for peer-review at EMBO Molecular Medicine. Should you be interested in having your work considered at our sister journal, please use the link below to transfer it. Roberto will then swiftly seek advice on your work from an expert in the field.

Thank you for your interest in our journal. I am very sorry to disappoint you on this occasion and I hope for the rapid publication of your study somewhere else.

Transferred to EMBO Molecular Medicine

18 May 2017

2nd Editorial Decision

09 June 2017

Thank you for the submission of your manuscript to EMBO Molecular Medicine. We have now heard back from the Reviewers whom we asked to evaluate your manuscript.

In aggregate, their thorough, thoughtful and strikingly convergent evaluations paint a clear picture: They appear to agree that study clearly has merits but much would need to be done to fully support the main conclusions.

I will not dwell into much detail, as the comments are self-explanatory. However, among the many important issues raised I would mention 1) The unclear clinical relevance of the findings (a relevant aspect for EMBO Molecular Medicine) due to the lack of analysis in terms of specific salivary gland types involved including with respect to human counterparts; 2) Lack of understanding of the process of innervation; 3) Undefined mechanism for the repopulation of the acinar population.

After our reviewer cross-commenting exercise and further editorial discussion, it was agreed that a revision would have to address all these issues. On the other hand, the concern emerged that such a revision might not be achievable within an acceptable timeframe, as it would involve long-term *in vivo* analysis. Furthermore, there is no guarantee of ultimate success.

However, given the potential interest of your findings and after internal discussion, we have decided to give you the opportunity to address the above issues.

We are thus prepared to consider a substantially revised submission, with the understanding that the Reviewers' concerns must be addressed with additional experimental data where appropriate as mentioned above and that acceptance of the manuscript will entail a second round of review. Although we would normally consider a period of 3-4 months to be appropriate for a revision, we are willing to grant more time if necessary. As for the mechanism of innervation, I will not be requiring you to perform these experiments (provided all other issues are carefully and fully dealt with); I do, however encourage you to develop your study as far as realistically possible in a mechanistic sense for your next, revised version to strengthen your findings and increase their impact.

Since the required revision in this case appears to require a significant amount of time, additional work and experimentation and might be technically challenging, I would therefore understand if you chose to rather seek publication elsewhere at this stage. Should you do so and although we hope not, we would appreciate a message to this effect. It would also be appreciated if you could provide a tentative timeline if you decide to pursue a revision. Please note that it is EMBO Molecular Medicine policy to allow a single round of revision only and that, therefore, acceptance or rejection of the manuscript will depend on the completeness of your responses included in the next, final version of the manuscript.

***** Reviewer's comments *****

Referee #1 (Remarks):

General comments;

The manuscript entitled, "Salivary glands regenerate after radiation injury through SOX2-mediated secretory cell replacement", by Emmerson E et al. demonstrated that irradiation-induced damaged acinar cells of salivary gland were recovered by Sox2+ stem cells essential to acinar cell maintenance. In addition, the authors showed that salivary glands could regenerate through a SOX2-nerve dependent mechanism using explant culture of human salivary with murine parasympathetic nerve. Indeed, the present study is of interest, but there are important points to be resolved as below.

Major comments

1. In the Material and Methods section, TG mice including Rosa26mTmG should be described more

in detail. Schematic diagram of the construction makes it easier to understand this system.

2. In Figure 1B, the authors showed the distribution of Sox2-expressing cells only in sublingual glands. Distribution of Sox2-expressing cells in submandibular glands should be shown compared with human SMG.

3. In Figure 1E, both AQP5 and MUC19 were detected as red fluorescence. Is there any possibility that red fluorescence of tdTomato fluorescent protein leaked in? GFP-positive cells were distributed in acinar cells 30 days after tamoxifen injection. It is interesting to show the distribution of SOX2+ cells at that time because author mentioned SOX2+ cells self-renew. In addition, is it possible that GFP+ cells replace most of acini at a longer time point? How long do acinar cells turn over?

4. In Figure 1, the authors mentioned that only 20% of AQP5+ acinar cells were SOX2+ cells. However, in Figure 2, a severe reduction in acinar cells of Sox2CreERT2RosaDTA mice was found after 7 days. The authors should explain this discrepancy. Is it possible that ablation of Sox2+ acinar cells affect maintenance of MUC19+ differentiated cells?

5. In Figure 4, SOX2+ cells in the IR gland replaced acinar cells. It is surprising that GFP+ cells replaced most of damaged acinar cells only after 48 hours. This seems to be inconsistent with previous reports, as the authors mentioned. If so, damaged acinar cells should die. The authors need to examine the cell death by TUNEL assay etc. Furthermore, a proliferative activity of SOX2+ cells should increase. The authors should examine a proliferative activity of SOX2+ cells by double immunostaining of SOX2 and Ki-67 (EdU, BrdU etc).

6. In Figure 4, it is difficult to compare the acinar cell repopulation by immunofluorescent analysis. Therefore, additional data such as a proliferative index is helpful to clearly understand this. Moreover, the authors suggested that IR activated SOX2+ cells repopulate tissue due to cholinergic signaling. It is necessary to show that Sox2+ cells have cholinergic receptors, such as M1 and M3.

7. In Figure 5, was human explant salivary gland obtained from irradiated patients? Please make sure this.

8. In Figure 5, interestingly, the authors demonstrated that nerves migrated in and around the human explant tissue. What factors induced nerve migration? Please make a comment about this in the Discussion part. Furthermore, it was shown that nerves actively maintained explant tissue structure. It is interesting whether cholinergic receptor antagonists, such as atropine and 4-DAMP, inhibit this.

9. It is unclear how Sox2+ cells with cholinergic stimulation proliferate and repopulate acinar cells, because cholinergic stimulation seems not to be directly related to cell proliferation and lineage commitment of acinar cells. Please mention more in detail in the Discussion.

Referee #2 (Remarks):

This manuscript investigates the potential of Sox2+ cells in the salivary gland to serve as a source of secretory acinar cells after irradiation. In addition, this report uncovers a link between parasympathetic innervation and Sox2+ progenitor cells, introduces the idea that radiation damage is due to loss of nerves and Sox2 progenitors, and demonstrates that a muscarinic mimetic can drive acinar cell expansion. This is an interesting paper, and demonstration that there are salivary gland stem cells that replace acinar cells would be a significant finding. But the data as presented do not wholly support the conclusions. There are several important issues that require further attention for this story to be compelling.

1. The authors seem to downplay the differences between submandibular, sublingual and parotid glands in order to broaden the interest for more than a specialist audience. However, this is misleading since the glands are not equivalent and are composed of different cell types and also show remarkably different sensitivity to radiation treatment, which is the basis for needing a stem cell. Throughout the manuscript, specific clarification of which salivary glands express Sox2 in mouse and human is missing. Arnold et al. (2011) previously showed that adult Sox2+ stem cells are

in murine sublingual gland (SL) (Note that this correction should be included in the first paragraph), but not in murine submandibular gland (SM). They did not mention parotid gland. Here, the authors demonstrate that human SM also has Sox2⁺ cells. Sox2 expression in human parotid is mentioned, but it is not clear if it is shown, since throughout the manuscript, it is often not clear which gland is shown or analyzed. (For example, Fig. 1D is labeled only as "salivary gland" and whether human, mouse, SL, SM, or parotid is not clear.) This is important, since Fig. 1E and EV1B show lineage traced adult Sox2 cells (presumably mouse), which would conflict with the results reported by Arnold et al. (2011), unless this is SL.

2. The authors suggest that the Sox2⁺ cells in murine SL are undifferentiated, because they do not express the differentiation marker mucin 19, but it is important to rule out that these cells are not the serous demilune cell type that is also prevalent in the SL. Serous demilune cells do not express mucin 19, but do stain prominently for alpha-amylase (Yamagashi et al. 2014 Acta Histochem Cytochem) and for lysozyme (Lee, Lim et al. 1990 Acta Histochem), both differentiated cell markers. To definitively prove that they are stem cells, it is critical to show that Sox2 cells do not exhibit any differentiated cell types.

3. Related to this, the authors clearly show that Sox2 expression in human SM is in acinar cells. This confuses the issue - are the Sox2⁺ cells undifferentiated stem cells, or are they acinar cells? The data shown in Fig. 2 confirm that elimination of Sox2⁺ cells severely depleted Aqp5⁺ acinar cells. But the severe reduction in acini (~75%; Fig. 2D) is not consistent with the number of cells expressing Sox2 (~21%; Fig. 1C). Is Sox2 activated in additional cells under stress? A major weakness is that the effects of ablation were analyzed at only one time point - 7 days after DTA induction - regeneration after ductal ligation takes up to 2 weeks, and regeneration after denervation and irradiation in later experiments was followed for up to 30 days. Why was this not done with the DTA ablation? Fig. EV2 states that ablation reduces acinar cell replacement - but acinar cell replacement was not measured. Thus, the statement that "regeneration is not possible without Sox2⁺ cells" has not been rigorously tested, or definitively proved.

4. The ablation shown in Fig. 2 appears to have been done in Sox2CreERT2Rosa26DTA; Rosa26mTmG/+ mice. If this is the case, why haven't the authors shown staining for GFP, to rule out that Sox2⁻ acinar cells have not been ablated? Further, why did they not look at lineage tracing after the ablation - to confirm that regeneration does not take place? To prove that Sox2 is required to generate new acinar cells why haven't they done a lineage tracing after deleting Sox2 transcription factor?

5. It is a likely possibility that the observed "self-duplication" of acinar cells (Aure et al. 2015) was due to a small population of proliferative progenitor cells. However, in order to disprove the report that acinar cells are replaced through cell duplication, the authors must show that the Sox2⁻ expressing cells do not include differentiated cells, and that Sox2 is not activated in those cells under stress. It is also necessary to show that the only cells proliferating are the Sox2⁺ cells. As the data stands now, the production of both Sox2⁺GFP⁺ and Sox2⁻ progeny (Fig. 4C) could be interpreted as consistent with acini self-duplication. This should be ruled out to support the assertion that "acini do not arise from self-duplication".

6. Fig. 3 shows a link between innervation and Sox 2 progenitors that could be better characterized. Does apoptosis increase after transection? Or are genes directly down-regulated? Expression levels of Sox2 decrease, but does the number of Sox2-expressing cells also change? (this could be readily checked in the Sox2EGFP mouse.) Is the lower number of regenerated acinar cells due to less Sox2 progenitors or to loss of cholinergic signal? This is important because comparing the photos in Fig. 3F assumes that they started with the same Sox2⁺ cell number for regeneration.

7. The conclusion that IR does not cause changes in innervation (Fig. 4) may have missed the window. Loss of innervation could occur rapidly after IR, and may be missed altogether if, as the authors point out, the plasticity in the peripheral nervous system leads to reappearance of nerves within 30 days after transection.

8. A more rigorous proof that cholinergic nerves play a role in controlling Sox2-mediated acinar cell replacement would be to activate the Sox2CreER after the injury - rather than prior. How long does it take for nerves in the transfected animals to disappear? During this interval, Sox2 progenitors

could be generating progeny. The results indicate that replacement is decreased after nerve injury - but do not directly test if nerves play a controlling role in Sox2 progenitor function. In fact, if Cre activation were done after the injury, the difference between injured and control might be more pronounced. Has cell proliferation been measured? Cell loss due to apoptosis?

9. There is some (presumably non-significant) increase in levels of Sox2 after IR at day 14. What about directly after IR? Fig. 4E - what are levels of Sox2 at 48 hours after IR in these cultures? It is important to rule out that Sox2 is induced by IR in random acinar cells. To address whether Sox2 progenitors are required for regeneration, IR should be done on the ablated glands, or in the Sox2 fl/fl mice after removal of Sox2 allele. Do cultures from Sox2-CreER;Sox2 fl/fl show less capacity for cell replacement after IR? When was Sox2CreER induced relative to the IR? If induced before, the experiment is not strictly testing the ability of Sox2 cells to regenerate the gland.

10. There are large numbers of Sox2+ cells in ex vivo explants - what is the level of cell proliferation? Does carbachol increase cell proliferation, or only differentiation of Sox2 progenitors? (Fig. EV4)

11. Were the ex vivo experiments shown in Fig. 5B-D done with healthy non-IR tissue or with IR-treated tissue? This is not clearly specified. The data show that culture with parasympathetic nerves can increase Sox2, and other markers - does it increase proliferation? No conclusions regarding proliferation (Fig. 5C) are stated. Were explants only from SM or also from parotid samples?

12. The authors have included an unrelated experiment in the manuscript showing short-term (14 day) lineage tracing of cKit+ cells in the mouse. Based on the rate of cell turnover in the salivary gland, this is an incomplete experiment, and should be removed - or expanded to include longer tracing times. These data do not contribute to the Sox2 story.

Additional points:

13. Were the 2x5Gy doses sequential, different days? Whole head/neck - where was dose rate of 5 Gy calculated? the surface of the mouse? Or the dose midway through the neck?

14. Is the GFP shown in most figures imaging direct fluorescence, or obtained through staining with GFP antibody? This should be included in Methods section.

15. Not specified: Are values presented in Fig. 5E averaged from samples incubated with 2 concentrations of CCh (100 and 200 nM)?

16. Reference to Fig. EV5A on page 10, should be corrected to EV4.

17. Labeling of all figures should be modified to indicate that GFP is derived from Sox2CreER mTmG and not from Sox2CreERGFP (which suggests a GFP insert linked to the CRE).

Referee #3 (Comments on Novelty/Model System):

Human tissue is used to identify Sox2 cells and mouse models are used to support this data. The gland examined in mouse is not the same gland as the human and this difference is not addressed. Additionally, IR model in mouse does not reflect human findings and differences are not adequately addressed.

Referee #3 (Remarks):

In this manuscript, the authors have identified a Sox2-positive population of cells in the submandibular salivary glands of humans that is capable of generating acini in mouse sublingual glands. During homeostasis, loss of Sox2 leads to decreased AQP5 and increased ducts in the sublingual gland of mice. The authors demonstrate that this Sox2-positive population is positively maintained by parasympathetic innervation in the mouse and imply that this mechanism may be conserved in humans. The authors suggest that human glands do not regenerate due to loss of Sox2+ progenitor cells and the nerves that regulate them. Reintroduction of innervation to the human tissues with CCh stimulation in culture is shown here as a novel co-culture assay. The findings are interesting and likely have relevance to human disease; however, there are significant issues that should be addressed.

Major points:

1. Only the sublingual gland (SL) is examined in mice in this study and so that authors should explicitly state that the study uses SL in the abstract and elsewhere in the manuscript.
2. It is unclear why SL mouse glands are compared with submandibular and parotid but not SL human.
3. Sox2 is a known potency marker. Are these cells positive for other known potency markers- eg. Oct4, Klf4, or Nanog?
4. Since neither Sox2⁺ cells are eliminated with genetic manipulation of mice, nor are acini, it is possible that there are other mechanisms for regulation of repopulation of the acinar population, including as the authors state does not occur, control through Mist1. Since the authors have not tested this directly, and have not costained for this marker, their conclusion that Sox 2⁺ cells are the only progenitor cells and that the Mist1 population does not contribute to tissue restoration is a gross overstatement. Interestingly, the authors report increased Mist1 transcripts in human tissue culture co-culture assays and in assays with CCh stimulation and should comment on this.
5. Fig EV1C In order to demonstrate that kit⁺ cells contribute to intercalated ducts, they should be identified with a marker.
6. Additionally, a lineage commitment marker for kit⁺ cells in the submandibular gland needs to be shown to make the statement that kit⁺ cells do or do not contribute to acinar cells in this gland.
7. Fig 2/FigEV2 Use of Sox2 fl/fl mouse does not appear to result in a severe lack of Sox2GFP area (Fig2A) as the graph in Fig 2C suggests, nor does there seem to be any significant change in innervation (Fig EV2). Use of the Sox2 DTA mouse only, however, appears to result in a phenotype. Fig EV2 Sox2 staining appears to be reduced in the Sox2 fl/fl mouse and not the Sox2 DTA mouse images. Please quantify the Sox2-positive nuclei in these mice to indicate loss of Sox2.
8. It is troubling throughout the figures that only three mice are included in calculations and yet statistics are shown. Is this because three samples from three mice are quantified to make n = 9? For example, Fig 2C-D It appears that 5 mice of each genotype were obtained and yet only 3 sections from each of 3 mice were used for quantification.
9. Fig3 Staining for decreased Muc19 is reported in the text but is not shown in the figure.
10. Fig 3B Since changes in Mist1 were observed with human tissue coculture and CCh stimulation in Fig5, please also report changes (if any) in Mist1 in this figure with transection.
11. It should be explained why Chrm1 and Chrm3 have not changed after 7 days if innervation has been removed. Additionally, its unclear why Tubb3 and Vip have increased beyond baseline levels.
12. In Fig 3C, the authors report a selective reduction in parasympathetic innervation. To make the conclusion, a parasympathetic nerve marker should be used to demonstrate a decrease with a sympathetic marker that does not decrease. Tubb3 used in this figure marks all nerves and is not a selective marker for parasympathetic innervation and does not appear to be decreased. Use of a more conspicuous color or single channel image and/or quantitative data are required.
13. Previously published literature in multiple organs, including salivary, indicates that IR induces tissue damage. The authors should include a marker to demonstrate efficacy of the IR in inducing a "genotoxic effect" - perhaps increased levels of proteins involved in DNA repair pathways or other previously reported changes characterized with IR.
14. Fig 4E To show that Sox2 is increased with IR at 48 hours in culture after IR, please quantify fluorescent areas for GFP and show representative cropped images of fluorescent areas.
15. In Fig 5A the change in Sox2 is not significant ($p < 0.05$) and should be reported as a trend.

16. To suggest decreased parasympathetic nerve function in human tissues, decreased levels of markers for parasympathetic nerve function should be shown as for the mouse model (eg. Vip, Vacht). As the authors state, Chrm markers are also present on acini and acini reported to be reduced, decreased Chrm1 and decreased Chrm3 may be due to decreased epithelium.

17. Fig5 C Ki67 staining appears higher in the mesenchyme only co-culture than with nerve co-culture, and ECAD appears to be reduced. Since ECAD levels appear to be significantly different in this assay, and the genes examined are epithelial, it's important to normalize to epithelium in addition to mesenchyme to account for this difference. Alternatively, demonstrate that there is no significant difference in the amount of epithelium present and include more representative images.

18. Fig5 D KIT and EGFR results indicate one sample of two at baseline, not increased. Myoepithelial/ductal markers also appear to be at or near baseline. Please correct text to reflect results reported.

19. Fig5 E/Fig EV5B Please clarify if human samples from different glands are averaged and if Fig 5/Fig EV5 are parts of the same data set as Fig. 5D. Methods/legend/ and results are unclear for Fig 5E/D. Fig EV5B indicates variability in the dataset that should be discussed. Additionally, if Fig. 5 is a summation of EV5 results, please indicate this.

Minor points:

1. DAPI and other markers shown blue are written in too dark of a color to show up in all of the figures. Also the use of blue in the 3 color overlays without single panel images makes it very difficult to discern the blue color. The authors should consider a color closer to cyan than blue.
2. Kwak, Alston et al., 2016 does not use K5 as a marker in this study as incorrectly stated bottom of page 6/top of page 7
3. Fig. 2. DAPI in bottom right panel of Fig 2A and Fig 2B appears to be brighter than other images. Are the exposures matched?
4. Fig EV3A Why is KRT5 shown and not referred to? What is the relevance of this marker to the model?
5. Fig EV4 is reported incorrectly in the text as EV5A.

1st Revision - authors' response

14 November 2017

We thank the reviewers for their very helpful and constructive criticism of our initial submission entitled "Salivary glands regenerate after radiation injury through SOX2-mediated secretory cell replacement". We have addressed all comments and substantiated our conclusions with extensive additional data. We have also added additional data into the manuscript to make our conclusions more clinically relevant. New data include:

1. Experiments demonstrating that the muscarinic receptors CHRM1 and 3 are expressed by SOX2+ cells and that *in vivo* administration of the muscarinic agonist pilocarpine promotes SOX2+ cell proliferation.
2. Genetic lineage tracing of SOX2+ cells for 30 days and KIT+ cells for 6 months showing extensive repopulation of the acinar compartment by SOX2+ cells and no acinar cell replenishment by KIT+ cells.
3. Extensive characterisation of the effects of radiation on the tissue over 30 days.
4. *In vivo* radiation experiments using the *Sox2^{CreERT2}; Sox2^{fl/fl}* mouse to demonstrate that SOX2+ progenitors are required for regeneration.
5. Short term ablation studies to demonstrate the gradual loss of tissue integrity and that glands do not ectopically express SOX2 as a result of damage/stress.
6. Genetic lineage tracing of SOX2+ cells in denervated glands where tamoxifen is activated before or after denervation.
7. *In vivo* denervation experiments in the Sox2eGFP mouse to directly determine the impact of denervation of SOX2+ cells.

We have addressed all the reviewers' comments below: reviewers comments are in black, our response is in red, and the updates to the text of the manuscript are in blue.

Referee #1 (Remarks):

General comments;

The manuscript entitled, "Salivary glands regenerate after radiation injury through SOX2-mediated secretory cell replacement", by Emmerson E et al. demonstrated that irradiation-induced damaged acinar cells of salivary gland were recovered by Sox2⁺ stem cells essential to acinar cell maintenance. In addition, the authors showed that salivary glands could regenerate through a SOX2-nerve dependent mechanism using explant culture of human salivary with murine parasympathetic nerve. Indeed, the present study is of interest, but there are important points to be resolved as below.

Major comments :

1. In the Material and Methods section, TG mice including Rosa26mTmG should be described more in detail. Schematic diagram of the construction makes it easier to understand this system.

We thank the reviewer for this comment and we have added some explanation into the methods section (1), the relevant part of the results section (2) and a schematic into EV1C, as follows:

- 1) "The *Rosa26^{mTmG}* mouse provides a valuable tool to lineage trace in conjunction with cell ablation/recombination (Muzumdar, Tasic et al., 2007). In brief, this model consists of a double-fluorescent Cre reporter that expresses membrane-targeted tandem dimer Tomato (mT) prior to Cre-mediated excision and membrane-targeted green fluorescent protein (mG) after excision (Fig. EV1D)."
- 2) "To determine whether SOX2⁺ cells contributed to acinar and duct lineages, we performed genetic lineage tracing using *Sox2^{CreERT2}* mice (Arnold et al, 2011) crossed to a *Rosa26^{mTmG}* reporter strain. The *Rosa26^{mTmG}* mouse is a double-fluorescent reporter mouse which when crossed with a Cre line expresses membrane-targeted tandem dimer Tomato (mT) prior to Cre-mediated excision and membrane-targeted green fluorescent protein (mG) after excision (Muzumdar et al, 2007) (Fig. EV1D). Thus, lineage traced cells will express mG."

2. In Figure1B, the authors showed the distribution of Sox2-expressing cells only in sublingual glands. Distribution of Sox2-expressing cells in submandibular glands should be shown compared with human SMG.

SOX2 is restricted to the adult SLG gland only in the mouse. SOX2 is absent in the murine SMG and PG. We have now included images in Figure 1 and Figure EV1 to demonstrate this. In contrast, SOX2 is expressed in all three of the major human glands (SMG, SLG, PG) and we have included images for all three in Figure 1. We have now clarified this in the text:

"We found SOX2 to be expressed by a subset of acinar cells in all three of the major adult human salivary glands (Fig. 1A, submandibular gland (SMG), sublingual gland (SLG), parotid gland (PG)). In the mouse, SOX2 protein was restricted to the adult murine SLG (absent from the SMG and PG, Fig. 1B and Fig. EV1A) where it was expressed by undifferentiated aquaporin (AQP)5+mucin (MUC)19-negative acinar cells (21% ± 4% of all AQP5+ acinar cells; Fig. 1C and D). Consistent with their potential role as a progenitor cell, ~6% of SOX2+AQP5+ cells co-expressed Ki67 (Fig. 1E and Fig. EV1B) while 19±4% were in the cell cycle (CyclinD1+; Fig. EV1C)."

3. In Figure1E, both AQP5 and MUC19 were detected as red fluorescence. Is there any possibility that red fluorescence of tdTomato fluorescent protein leaked in? GFP-positive cells were distributed in acinar cells 30 days after tamoxifen injection. It is interesting to show the distribution of SOX2+ cells at that time because author mentioned SOX2+ cells self-renew. In addition, is it possible that GFP+ cells replace most of acini at a longer time point? How long do acinar cells turn over?

We apologize for the confusion in this figure. In this assay AQP5 and MUC19 were immunostained using far red (Cy5; 633nm) and not red (Cy3 or mT; 594nm). However, in our images the red mT excitation wavelength was omitted and the far red excitation outcome is shown in red for clarity. We have clarified this in the text:

"It should be noted that in Fig. 1E we omitted the membrane-bound tomato (mT) signal for clarity and AQP5 and MUC19 are immunolabeled using Cy5-conjugated secondary antibody (633nm)."

The 30 day time point in Fig. 1E (now Fig. 1F) was chosen to show that acinar cells are replaced by SOX2+ cells by 30 days. To demonstrate this, we have added an image for SOX2 lineage tracing at 30 days, in addition to 14 days (see Fig. EV1E).

We have tried a number of methods to determine how quickly acinar cells turn over, with no success. However, the extent of lineage tracing that we see with the *Sox2^{CreERT2}; Rosa26^{mTmG}* mouse after 14 days (~50% of acinar cells) and 30 days (>95% of acinar cells) suggests that the majority of acinar cells have been replaced between 14 and 30 days (Fig EV1E).

4. In Figure 1, the authors mentioned that only 20% of AQP5+ acinar cells were SOX2+ cells. However, in Figure 2, a severe reduction in acinar cells of *Sox2CreERT2RosaDTA* mice was found after 7 days. The authors should explain this discrepancy. Is it possible that ablation of Sox2+ acinar cells affect maintenance of MUC19+ differentiated cells?

We thank the reviewer for this comment and we have clarified the severe reduction in AQP5+ cells following SOX2+ cell ablation in better detail in the text as follows:

“Furthermore, ablation of SOX2+ cells resulted in few remaining acini by 8 days, as shown by large regions of the ductal network completely devoid of AQP5+ cells (ducts are marked by dotted lines or KRT8 in Fig. 2B) (Fig. 2B, D, Fig EV2A). To exclude the possibility that tissue degeneration was solely due to destabilization of the tissue rather than loss of acinar cell replacement, we examined SLG after a short-term ablation. As shown in Figure EV3A and 3B, at day 4 or 5 (3 or 4 days of tamoxifen treatment), few SOX2+ cells remained in the gland in both the *Sox2^{CreERT2}*; *Sox2^{fl/fl}* and *Sox2^{CreERT2}*; *Rosa26^{DTA}* SLG (Fig. EV3A, B) and *Sox2* transcripts were substantially reduced (Fig. EV3B), however, acini were present albeit disorganized and atrophic in appearance.”

5. In Figure 4, SOX2+ cells in the IR gland replaced acinar cells. It is surprising that GFP+ cells replaced most of damaged acinar cells only after 48 hours. This seems to be inconsistent with previous reports, as the authors mentioned. If so, damaged acinar cells should die. The author need to examine the cell death by TUNEL assay etc. Furthermore, a proliferative activity of SOX2+ cells should increase. The authors should examine a proliferative activity of SOX2+ cells by double immunostaining of SOX2 and Ki-67 (EdU, BrdU etc).

We have now included quantification data on SOX2+ cells and cycling SOX2+ cells (CCND1+SOX2+ cells) following irradiation (Fig. 5C). Here we demonstrate that immediately following IR (Day 1) there is a significant reduction in SOX2+ cells which returns to control levels by day 3. Concurrently, the number of cycling SOX2+ cells increases significantly at days 3 and 7 post-IR, suggesting that IR stimulates SOX2+ cell proliferation, resulting in extensive SOX2-mediated acinar cell replacement by day 14 (in vivo) and 48h (ex vivo).

6. In Figure 4, it is difficult to compare the acinar cell repopulation by immunofluorescent analysis. Therefore, additional data such as a proliferative index is helpful to clearly understand this. Moreover, the authors suggested that IR activated SOX2+ cells repopulate tissue due to cholinergic signaling. It is necessary to show that Sox2+ cells have cholinergic receptors, such as M1 and M3.

We thank the reviewer for this suggestion and we have now included flow cytometry data to demonstrate that 95% of epithelial SOX2+ (EpCAM+SOX2+) SLG cells express CHRM1 and 99% express CHRM3. This strengthens the conclusion that SOX2 cells are regulated by cholinergic signals.

We have tried a number of methods to determine the proliferative index of acinar SOX2+ cells with no success. However, the extent of lineage tracing that we see with the *Sox2^{CreERT2}; Rosa26^{mTmG}* mouse after 30 days (>95%) suggests that within 30 days the majority of acinar cells have been replaced. We have also now quantified the number of cycling SOX2+ (SOX2+CCND1+) cells over a timecourse of IR in Fig. 5C and show that following an initial reduction in the number of SOX2+ cells immediately following IR (day 1) SOX2+ cells become more proliferative at Days 3 and 7 (CCND1+), which contributes to the SOX2-mediated acinar cell replacement we observe following IR injury.

7. In Figure 5, was human explant salivary gland obtained from irradiated patients? Please make sure this.

We apologise for this lack of information. The tissue used in this assay was from non-IR patients. We have clarified this in the text as follows:

“In this model, non-irradiated human SMG is dissected into <1mm pieces and placed alongside an embryonic day 13 murine submandibular gland parasympathetic ganglia (contains mesenchyme) or mesenchyme only (control, no nerves).”

8. In Figure 5, interestingly, the authors demonstrated that nerves migrated in and around the human explant tissue. What factors induced nerve migration? Please make a comment about this in the Discussion part. Furthermore, it was shown that nerves actively maintained explant tissue structure. It is interesting whether cholinergic receptor antagonists, such as atropine and 4-DAMP, inhibit this.

We performed additional qPCR for the neuroattractants neurturin (NRTN) and nerve growth factor (NGF) and found no increase in expression in the presence of the ganglia compared to the mesenchyme alone (Figure 5D). This suggests that the epithelia produces these neuroattractants whether the nerves are present or not (i.e. the mesenchyme is supportive enough to maintain epithelial homeostasis) or that this phenomenon occurs earlier in the culture (i.e. at an earlier stage when the nerves are just starting to envelop the epithelia) and any differences have been resolved by this later time point (day 7). We have incorporated this into the text as follows:

“We analysed the explants for the neuroattractants neurturin (*NRTN*) and nerve growth factor (*NGF*) (Knox et al. 2013) in an attempt to ascertain what factors induced this nerve migration. However, we found no increase in expression in the presence of the ganglia compared to the mesenchyme alone (Fig. 6D), suggesting that other neuroattractants were being synthesized, that the epithelia produces these neuroattractants whether the nerves are present or not (i.e. the mesenchyme is supportive enough to maintain epithelial homeostasis) or that this phenomenon occurs earlier in the culture (i.e. at an earlier stage when the nerves are just starting to envelop the epithelia) and any differences have been resolved by this later time point (day 7).”

9. It is unclear how Sox2+ cells with cholinergic stimulation proliferate and repopulate acinar cells, because cholinergic stimulation seems not to be directly related to cell proliferation and lineage commitment of acinar cells. Please mention more in detail in the Discussion.

We thank the reviewer for this comment. We have performed additional experiments where we treated mice with pilocarpine, a cholinergic agonist, and demonstrate a substantial increase in the number of proliferating SOX2+ cells (co-stained with Ki67) compared with saline control. Thus, we conclude that cholinergic stimulation induces SOX2+ cell proliferation, which leads to repopulation of acinar cells during homeostasis and following injury.

“To ensure that SOX2+ cells of the SLG were capable of responding directly to acetylcholine produced by the parasympathetic nerves, we analysed expression of acetylcholine/muscarinic receptors by SOX2+ cells as well as their ability to respond to muscarinic agonists *in vivo*. SOX2+ cells expressed both CHRM1 and CHRM3 (95% and 99% respectively; Fig. 4A, B) and short term treatment of wild type mice with the muscarinic agonist pilocarpine (delivered I.P. and sacrificed at 18h) increased the number of proliferating SOX2+ cells (SOX2+Ki67+ cells; Fig. 4C, D). The total number of SOX2+ cells, however, was not significantly changed by 18h post-injection indicating that muscarinic activation does not induce ectopic expression of SOX2. Thus, these data support our hypothesis that parasympathetic nerves via acetylcholine muscarinic signalling maintain SOX2+ progenitors and acini and promote acinar cell replenishment.”

We have also discussed this in the text of the discussion as follows:

“We further show that cholinergic nerves play a vital role in controlling SOX2-mediated acinar cell replacement and that this neuronal influence can be replicated through addition of cholinergic mimetics to irradiated tissue. By treating mice *in vivo* with a cholinergic agonist we demonstrate that this effect appears to be mediated primarily via increased proliferation of SOX2+ cells, which in turn leads to repopulation of acinar cells during homeostasis and following injury.”

Referee #2 (Remarks):

This manuscript investigates the potential of Sox2⁺ cells in the salivary gland to serve as a source of secretory acinar cells after irradiation. In addition, this report uncovers a link between parasympathetic innervation and Sox2⁺ progenitor cells, introduces the idea that radiation damage is due to loss of nerves and Sox2 progenitors, and demonstrates that a muscarinic mimetic can drive acinar cell expansion. This is an interesting paper, and demonstration that there are salivary gland stem cells that replace acinar cells would be a significant finding. But the data as presented do not wholly support the conclusions. There are several important issues that require further attention for this story to be compelling.

1. The authors seem to downplay the differences between submandibular, sublingual and parotid glands in order to broaden the interest for more than a specialist audience. However, this is misleading since the glands are not equivalent and are composed of different cell types and also show remarkably different sensitivity to radiation treatment, which is the basis for needing a stem cell. Throughout the manuscript, specific clarification of which salivary glands express Sox2 in mouse and human is missing. Arnold et al. (2011) previously showed that adult Sox2⁺ stem cells are in murine sublingual gland (SL) (Note that this correction should be included in the first paragraph), but not in murine submandibular gland (SM). They did not mention parotid gland. Here, the authors demonstrate that human SM also has Sox2⁺ cells. Sox2 expression in human parotid is mentioned, but it is not clear if it is shown, since throughout the manuscript, it is often not clear which gland is shown or analyzed. (For example, Fig. 1D is labeled only as "salivary gland" and whether human, mouse, SL, SM, or parotid is not clear.) This is important, since Fig. 1E and EV1B show lineage traced adult Sox2 cells (presumably mouse), which would conflict with the results reported by Arnold et al. (2011), unless this is SL.

We apologize for this omission. As reported by Arnold et al. (2011) SOX2 is restricted to the adult SLG gland only in the mouse. SOX2 is absent in the murine SMG and PG. We have now included images in Figure 1 and Figure EV1 to demonstrate this. In contrast, SOX2 is expressed in all three of the major human glands (SMG, SLG, PG) and we have included images for all three in Figure 1. We have now clarified this in the text as follows:

“SOX2 has been established as a progenitor cell marker in the fetal mouse submandibular and sublingual salivary glands but whether SOX2⁺ cells in the adult tissue also produce acinar and duct cells is unclear (Arnold et al. 2011; Emmerson et al. 2017). Furthermore, whether these cells are also present in adult human salivary glands is not known. We found SOX2 to be expressed by a subset of acinar cells in all three of the major adult human salivary glands (Fig. 1A, submandibular gland (SMG), sublingual gland (SLG), parotid gland (PG)). In the mouse, SOX2 protein was restricted to the adult murine SLG (absent from the SMG and PG, Fig. 1B and Fig. EV1A) where it was expressed by undifferentiated aquaporin (AQP)5⁺ mucin (MUC)19-negative acinar cells (21% ± 4% of all AQP5⁺ acinar cells; Fig. 1C and D).”

We have clarified the nomenclature throughout the manuscript where from Figure 1C onwards we only use mouse SLG and refer to it as such.

2. The authors suggest that the Sox2⁺ cells in murine SL are undifferentiated, because they do not express the differentiation marker mucin 19, but it is important to rule out that these cells are not the serous demilune cell type that is also prevalent in the SL. Serous demilune cells do not express mucin 19, but do stain prominently for alpha-amylase (Yamagashi et al. 2014 Acta Histochem Cytochem) and for lysozyme (Lee, Lim et al. 1990 Acta Histochem), both differentiated cell markers. To definitively prove that they are stem cells, it is critical to show that Sox2 cells do not exhibit any differentiated cell types.

SOX2⁺ cells are not stem cells but undifferentiated lineage-restricted progenitor cells. They give rise to differentiated MUC19⁺ mucous acinar cells, but do not give rise to other cell populations of the salivary gland (such as ductal cells, myoepithelial cells, etc). We have clarified this in the text as follows:

“Thus, our lineage tracing results indicate that SOX2+ cells are lineage-restricted progenitor cells that give rise to differentiated progeny, similar to what has been observed in the epidermis, intestine and incisor (Barker, 2014; Owens & Watt, 2003; Seidel et al. 2017).”

3.1 Related to this, the authors clearly show that Sox2 expression in human SM is in acinar cells. This confuses the issue - are the Sox2+ cells undifferentiated stem cells, or are they acinar cells?

We apologize for the confusion. SOX2+ cells in both the murine SLG and the human SMG, SLG and PG are found in the acini, express acinar markers, including AQP5 and MIST1, but do not express differentiation markers, such as MUC19. Thus, they are a progenitor cell located in the acini. They can be both acinar (i.e. in the acinus) but be a progenitor cell and undifferentiated.

3.2 The data shown in Fig. 2 confirm that elimination of Sox2+ cells severely depleted Aqp5+ acinar cells. But the severe reduction in acini (~75%; Fig. 2D) is not consistent with the number of cells expressing Sox2 (~21%; Fig. 1C). Is Sox2 activated in additional cells under stress? A major weakness is that the effects of ablation were analyzed at only one time point - 7 days after DTA induction - regeneration after ductal ligation takes up to 2 weeks, and regeneration after denervation and irradiation in later experiments was followed for up to 30 days. Why was this not done with the DTA ablation? Fig. EV2 states that ablation reduces acinar cell replacement - but acinar cell replacement was not measured. Thus, the statement that "regeneration is not possible without Sox2+ cells" has not been rigorously tested, or definitively proved.

We apologize that the results of the ablation of SOX2+ cells are not clear. In this model we show that the ablation of SOX2+ cells leads to a substantial loss of AQP5+ acinar cells because once these SOX2+ progenitor cells are ablated, there are no cells to replace differentiated acinar cells. We have performed short term ablation of SOX2+ cells and have clarified the text as follows:

“Ablation of *Sox2* from SOX2+ cells or elimination of SOX2+ cells via DTA severely depleted SOX2+ and AQP5+ cells but not KRT8+ ductal cells indicating *Sox2* and SOX2+ cells were necessary for maintaining functional acini (Fig. 2A-D; efficiency of *Sox2* or SOX2+ cell ablation is shown in Fig. EV2A). In the absence of *Sox2*, acinar but not ductal cells exited the cell cycle, as shown by the decrease in cyclin D1 (CCND1)+ acinar cells (Fig. EV2D). Furthermore, ablation of SOX2+ cells resulted in few remaining acini by 8 days (Fig. 2B, D, Fig. EV2A), as shown by large regions of the ductal network completely devoid of AQP5+ cells (ducts are marked by dotted lines or KRT8 in Fig. 2B). To exclude the possibility that tissue degeneration was solely due to destabilization of the tissue rather than loss of acinar cell replacement, we examined SLG after a short-term ablation. As shown in Fig. EV3A and 3B, at day 4 or 5 (3 or 4 days of tamoxifen treatment), few SOX2+ cells remained in the gland of both the *Sox2^{CreERT2}*; *Sox2^{fl/fl}* and *Sox2^{CreERT2}*; *Rosa26^{DTA}* SLG (Fig. EV3A, B) and *Sox2* transcripts were substantially reduced (Fig. EV3B). However, acini were present albeit disorganized and atrophic in appearance. Furthermore, we did not observe an increase in SOX2+ cells (or *Sox2* transcripts), indicating that SOX2 is not ectopically expressed in acinar cells in response to tissue damage.”

We agree with the reviewer that it would be interesting to extend the timepoints used for the DTA experiments to see if regeneration occurred at a later stage following ablation of SOX2+ cells. However, there was no option to analyse regeneration past the point shown in this assay due to the poor health of the mice past 7 days (BCS<2 requires euthanasia according to our IACUC procedures). This is likely due to the loss of SOX2+ cells in other tissue systems (such as the oesophagus, forestomach and glandular stomach).

We have also now included experiments where we ablate Sox2 expression (*Sox2^{CreERT2}*; *Sox2^{fl/fl}* mice) and irradiate to induce SLG damage and found a dramatic loss of AQP5+ acinar cells and disrupted tissue architecture 14 days later in the absence of *Sox2* (Fig. EV6) compared with the extent that we see in wild-type mice 14 days post-IR (Fig. EV6). This further confirms that *Sox2* is essential for SLG regeneration following radiation injury. In addition, we find that other acinar cells do not switch on SOX2 in response to IR stress (Fig. EV6).

4. The ablation shown in Fig. 2 appears to have been done in **Sox2CreERT2Rosa26DTA; Rosa26mTmG/+ mice**. If this is the case, why haven't the authors shown staining for GFP, to rule out that Sox2- acinar cells have not been ablated? Further, why did they not look at lineage tracing

after the ablation - to confirm that regeneration does not take place? To prove that Sox2 is required to generate new acinar cells why haven't they done a lineage tracing after deleting Sox2 transcription factor?

We thank the reviewer for this comment. We have altered the channel colours we have used in Fig. 2 to make the figure easier to understand and in the lower panels we have included the Sox2CreERGF channel (in red) in both the Sox2fl/fl and Rosa DTA models to demonstrate that in the wild-type littermates there is no Cre-mediated deletion. In contrast in the *Sox2^{CreERT2}; Sox2^{fl/fl}* we see extensive GFP+ cells (in red) indicating where *Sox2* has been ablated from cells. However, in the case of the *Sox2^{CreERT2}; Rosa26^{DTA}* model Cre activation results in cell death (by use of the DTA) meaning we do not see any GFP+ cells (in red) as these cells have died. Thus, the fact that we have ablated SOX2+ cells and see such a striking loss of AQP5+ acinar cells shows we see little to no regeneration in the *Sox2^{CreERT2}; Rosa26^{DTA}* model.

5. It is a likely possibility that the observed "self-duplication" of acinar cells (Aure et al. 2015) was due to a small population of proliferative progenitor cells. However, in order to disprove the report that acinar cells are replaced through cell duplication, the authors must show that the Sox2-expressing cells do not include differentiated cells, and that Sox2 is not activated in those cells under stress. It is also necessary to show that the only cells proliferating are the Sox2+ cells. As the data stands now, the production of both Sox2+GFP+ and Sox2-negative progeny (Fig. 4C) could be interpreted as consistent with acini self-duplication. This should be ruled out to support the assertion that "acini do not arise from self-duplication".

SOX2+ cells in the murine SLG are found in the acini, express acinar markers, including AQP5 and MIST1, but do not express differentiation markers, such as MUC19. Thus, they are a progenitor cell located in the acini. They are acinar progenitor cells and do not give rise to other cell types (such as ductal cells, myoepithelial cells), but are lineage committed and give rise to mature acinar cells (MUC19+, see Fig 1C).

We do not expect SOX2 cells to be the only proliferating cells. In the majority of organ systems e.g., such as the intestine, the incisor and epidermis transit amplifying cells are present and we expect the same for this organ. These cells will be derived from the SOX2+ cells, as shown by our lineage tracing data. We show the data (in Fig. EV1B) that while 5.6% of SOX2+ acinar cells are proliferating (Ki67+), 16.5% of SOX2- acinar cells are proliferating. Thus, SOX2+ acinar cells are not the only proliferating cells. We have included this in the text below. We have also re-worded the statement that "acini do not arise from self-duplication" to "acini do not arise from the self-duplication of fully differentiated acinar cells".

"Our lineage tracing analysis confirmed that SOX2+ cells give rise to acinar but not duct cells. However, as we also observed the presence of Ki67+SOX2- acinar cells (~6% SOX2+Ki67+ and 16.5% SOX2-Ki67+ cells, Fig. EV1B), suggestive of an alternative progenitor cell or a transit amplifying cell for the acinar lineage, we investigated the requirement of SOX2 and SOX2+ cells in SLG maintenance and repair by genetically removing *Sox2* in SOX2+ cells using *Sox2^{CreERT2}; Sox2^{fl/fl}* mice (Fig. 2A, C) or ablated SOX2+ cells using diphtheria toxin (DTA) expressed under the control of the inducible *Sox2* promoter (*Sox2^{CreERT2}; Rosa26^{DTA}*; Fig. 2B, D). In the latter assay, SOX2+ cells undergo cell death in response to intracellular production of DTA. Ablation of *Sox2* from SOX2+ cells or elimination of SOX2+ cells via DTA severely depleted SOX2+ and AQP5+ cells but not KRT8+ ductal cells indicating *Sox2* and SOX2+ cells were necessary for maintaining functional acini (Fig. 2A-D; efficiency of *Sox2* or SOX2+ cell ablation is shown in Fig. EV2A). In the absence of *Sox2*, acinar but not ductal cells exited the cell cycle, as shown by the decrease in cyclin D1 (CCND1)+ acinar cells (Fig. EV2D). Furthermore, ablation of SOX2+ cells resulted in few remaining acini by 8 days (Fig. 2B, D, Fig. EV2A), as shown by large regions of the ductal network completely devoid of AQP5+ cells (ducts are marked by dotted lines or KRT8 in Fig. 2B). To exclude the possibility that tissue degeneration was solely due to destabilization of the tissue rather than loss of acinar cell replacement, we examined SLG after a short-term ablation. As shown in Fig. EV3A and 3B, at day 4 or 5 (3 or 4 days of tamoxifen treatment), few SOX2+ cells remained in the gland of both the *Sox2^{CreERT2}; Sox2^{fl/fl}* and *Sox2^{CreERT2}; Rosa26^{DTA}* SLG (Fig. EV3A, B) and *Sox2* transcripts were substantially reduced (Fig. EV3B). However, acini were present albeit disorganized and atrophic in appearance. Furthermore, we did not observe an increase in SOX2+ cells (or *Sox2* transcripts), indicating that SOX2 is not ectopically expressed in acinar cells in

response to tissue damage. We also determined if alterations in tissue composition were due to reduced innervation, an essential regulator of tissue function. However, we measured similar innervation in *Sox2^{CreERT2}; Sox2^{fl/fl}* SLG to wild-type controls and a significant increase in axon bundles in *Sox2^{CreERT2}; Rosa26^{DTA}* SLG (Fig. EV2B and C). The latter finding suggests ablation of cells triggers the release of factors that promote innervation but that, even with increased innervation, regeneration is not possible without SOX2+ cells. In sum, these results indicate that SOX2+ cells, at least under the conditions tested, are the sole acinar progenitors in the SLG and that acini do not arise from the self-duplication of fully differentiated acinar cells, as suggested previously (Aure et al, 2015). Similar to studies in the epidermis, intestine and incisor (Barker, 2014; Owens & Watt, 2003; Seidel et al, 2017), our data also suggest the presence of a transit amplifying population derived from SOX2+ cells that may be involved in rapidly repopulating the acinar compartment.”

We agree with the reviewer that including experiments that show that that SOX2 is not activated in SOX2- cells under stress make an important addition. We show this is not the case both after DTA (see above text) and flox excision of Sox2. We have also now included data where *Sox2^{CreERT2}; Sox2^{fl/fl}* mice were irradiated (as per our wild-type IR experiments) and analyzed for SOX2+ cells (Fig. EV7). We find that other acinar cells do not switch on SOX2 in response to IR stress (Fig. EV7). We also find that in the absence of *Sox2* we see a dramatic loss of AQP5+ acinar cells and general tissue architecture following radiation, confirming that *Sox2* is essential for acinar cell replacement following IR injury.

6. Fig. 3 shows a link between innervation and Sox2 progenitors that could be better characterized. Does apoptosis increase after transection? Or are genes directly down-regulated? Expression levels of Sox2 decrease, but does the number of Sox2-expressing cells also change? (this could be readily checked in the Sox2EGFP mouse.) Is the lower number of regenerated acinar cells due to less Sox2 progenitors or to loss of cholinergic signal? This is important because comparing the photos in Fig. 3F assumes that they started with the same Sox2+ cell number for regeneration.

In Figure 3 we demonstrate that both *Sox2* transcript levels (Fig. 3B) and SOX2 protein (Fig. 3F and Fig. EV3C) are reduced following nerve transection. As suggested by the reviewer we have further analysed this using the *Sox2^{eGFP}* mouse and find that not only *Sox2* transcript and SOX2 protein, but the expression of GFP is reduced 7 days following nerve transection. We have now shown this new data in Fig. 3.

It is important to note that GFP is expressed only if the *Sox2* promoter is active such that we are unable to identify cells that previously expressed SOX2 (the GFP turns off). However, we do not find any significant difference in CASP3 staining (Fig. EV3G) or markers of cell death (EV3H) at the transcriptional level indicating that we do not lose SOX2+ cells to cell death but that the expression of SOX2 requires cholinergic signaling. We do see a significant reduction in expression of CyclinD1 (*Ccnd1*), suggesting that the defect may be due to cell cycle and proliferation. We have clarified our findings in the text as follows:

“Denervation resulted in reduced acinar cell size (as observed previously (Patterson et al, 1975); Fig. EV4B), decreased AQP5 protein and transcript levels of the differentiated acinar cell marker *Muc19* (Fig. 3B, D). Interestingly, transcript and protein levels of MIST1 were unchanged following denervation (Fig. 3B, E, F), suggesting that while functional markers of acinar cells are disrupted in the absence of innervation, acinar cell identity is not adversely affected. Strikingly, SOX2+ cells lose expression of *Sox2* (demonstrated using the *Sox2eGFP* mouse) and the levels of SOX2 protein and transcript were greatly reduced in the absence of innervation (Fig. 3B, C, F and Fig. EV4C), indicating SOX2 maintenance requires innervation. To determine whether SOX2+ cells remained capable of repopulating the tissue after denervation, we performed genetic lineage tracing where Cre driven by the endogenous *Sox2* promoter (*Sox2^{CreERT2}; Rosa26^{mTmG}*) was activated 3 days after denervation and traced until day 14. As shown in Figs. 3G and 3H, acinar cell replacement by SOX2+ progenitors was significantly reduced (~50%) 14 days after transection. Similarly, in SLG in which recombination was induced before nerve transection (tamoxifen one day prior to transection), acinar cell replacement by SOX2+ progenitors was significantly depleted (~50%) after 14 days (Fig. EV4E and F). Reduced acinar cell replacement is likely due to decreased cell proliferation rather than cell death as we measured a significant reduction in CyclinD1 (*Ccnd1*) while markers of cell death (activated caspase-3 (CASP3+) cells in Fig. EV4G or *Bax*, *Pmaip1* (NOXA), and *Bbc3*

(PUMA) in Fig. EV4H)) were either not observed or remained unchanged. The absence of cell death also suggests that cells previously positive for SOX2 continue to be present but that cholinergic innervation is essential for maintaining SOX2 expression.

In this experiment we transect the chorda tympani on one side of the mouse only, leaving the contralateral gland as an internal control. This is the most robust control since the contralateral gland matches biologically and genetically. In this respect there is no reason to believe that the control versus the experimental gland would differ in the number of SOX2+ cells prior to nerve transection. We have also quantified SOX2+ cells in a large number of glands and find very similar quantities of SOX2+ cells. For the lineage tracing model shown in 3H and Fig. EV3E, F we performed this in two ways: 1) inducing recombination on the day prior to nerve transection to label single cells (where the glands would start with comparable numbers of SOX2+ cells), and 2) inducing recombination 3 days after nerve transection, to ensure that there is no residual nerve signalling in the initial days of lineage tracing. Outcomes were highly similar with the extent of lineage tracing being reduced to approx. 50% in the transected CT compared to the intact control (see Fig 3H and Fig. EV3E, F). Thus, we are confident in our conclusion that neuronal signals maintain SOX2-mediated cell replacement.

7. The conclusion that IR does not cause changes in innervation (Fig. 4) may have missed the window. Loss of innervation could occur rapidly after IR, and may be missed altogether if, as the authors point out, the plasticity in the peripheral nervous system leads to reappearance of nerves within 30 days after transection.

We thank the reviewers for this comment. We have included extensive data looking at transcriptional changes of a timecourse (day 1, 3, 7, 14, 30) following IR. Here we show significant reduction in innervation (the pan-neuronal marker, *Tubb3*, and the parasympathetic-derived neuropeptide, *Vip*) 1 day post-IR, indicating that nerves are adversely affected in the early stages following IR damage but recover. In addition, we have now quantified the number of SOX2+ cells in a timecourse following IR and find that while there is a significant reduction in the number of SOX2+ cells at day 1 post-IR the numbers return to control levels by days 3 and 7 post-IR.

8. A more rigorous proof that cholinergic nerves play a role in controlling Sox2-mediated acinar cell replacement would be to activate the Sox2CreER after the injury - rather than prior. How long does it take for nerves in the transected animals to disappear? During this interval, Sox2 progenitors could be generating progeny. The results indicate that replacement is decreased after nerve injury - but do not directly test if nerves play a controlling role in Sox2 progenitor function. In fact, if Cre activation were done after the injury, the difference between injured and control might be more pronounced. Has cell proliferation been measured? Cell loss due to apoptosis?

We thank the reviewer for this comment. We ask the reviewer to see our response to point 6 and have placed the text from the manuscript that addresses the reviewers comments below.

“Denervation resulted in reduced acinar cell size (as observed previously (Patterson et al, 1975); Fig. EV4B), decreased AQP5 protein and transcript levels of the differentiated acinar cell marker *Muc19* (Fig. 3B, D). Interestingly, transcript and protein levels of MIST1 were unchanged following denervation (Fig. 3B, E, F), suggesting that while functional markers of acinar cells are disrupted in the absence of innervation, acinar cell identity is not adversely affected. Strikingly, SOX2+ cells lose expression of *Sox2* (demonstrated using the *Sox2eGFP* mouse) and the levels of SOX2 protein and transcript were greatly reduced in the absence of innervation (Fig. 3B, C, F and Fig. EV4C), indicating SOX2 maintenance requires innervation. To determine whether SOX2+ cells remained capable of repopulating the tissue after denervation, we performed genetic lineage tracing where Cre driven by the endogenous *Sox2* promoter (*Sox2^{CreERT2}; Rosa26^{mTmG}*) was activated 3 days after denervation and traced until day 14. As shown in Figs. 3G and 3H, acinar cell replacement by SOX2+ progenitors was significantly reduced (~50%) 14 days after transection. Similarly, in SLG in which recombination was induced before nerve transection (tamoxifen one day prior to transection), acinar cell replacement by SOX2+ progenitors was significantly depleted (~50%) after 14 days (Fig. EV4E and F). Reduced acinar cell replacement is likely due to decreased cell proliferation rather than cell death as we measured a significant reduction in CyclinD1 (*Ccnd1*) while markers of cell death (activated caspase-3 (CASP3+) cells in Fig. EV4G or *Bax*, *Pmaip1* (NOXA), and *Bbc3* (PUMA) in Fig. EV4H)) were either not observed or remained unchanged. The absence of cell death

also suggests that cells previously positive for SOX2 continue to be present but that cholinergic innervation is essential for maintaining SOX2 expression.

9. There is some (presumably non-significant) increase in levels of Sox2 after IR at day 14. What about directly after IR? Fig. 4E - what are levels of Sox2 at 48 hours after IR in these cultures? It is important to rule out that Sox2 is induced by IR in random acinar cells. To address whether Sox2 progenitors are required for regeneration, IR should be done on the ablated glands, or in the Sox2 fl/fl mice after removal of Sox2 allele. Do cultures from Sox2-CreER; Sox2 fl/fl show less capacity for cell replacement after IR? When was Sox2CreER induced relative to the IR? If induced before, the experiment is not strictly testing the ability of Sox2 cells to regenerate the gland.

We induce recombination 24h before IR in order to label single cells (which we show as a panel in Fig. 4E at 0h), which then go on to populate the tissue following IR damage. It takes at least 12 h for the tamoxifen to be metabolised by the liver and synthesized into the active form of hydroxytamoxifen and as such we can't do this assay in another order as we'd potentially miss any very early regeneration/cell replacement following IR if we gave the TAM at the same time or later. We have clarified this in the text as follows:

“This time point was chosen as a lag time of 12-24 hours has been previously reported for tamoxifen-induced recombination of Cre lines in mice (Nakumura et al). Thus, single SOX2+ cells have been labelled by 24h after injection (Fig. 4E).”

We have also included a schematic of TAM administration timing in Fig. 4.

We have also included a more extensive characterization of gene expression following in vivo irradiation where we demonstrate that expression of *Sox2* is significantly downregulated at days 1 and 3 post-IR, but returns to control levels by day 7 (see fig. EV5B). We have also now quantified the number of cycling SOX2+ (SOX2+CCND1+) cells over a timecourse of IR in Fig. 5C and show that following an initial reduction in the number of SOX2+ cells immediately following IR (day 1) SOX2+ cells become more proliferative at Days 3 and 7 (CCND1+), which contributes to the SOX2-mediated acinar cell replacement we observe following IR injury.

We agree with the reviewer that IR experiments following *Sox2* ablation make an important addition and we have now included data where *Sox2*^{CreERT2}; *Sox2*^{fl/fl} mice were irradiated (as per our wild-type IR experiments) and analyzed for SOX2+ cells and AQP5+ acinar cells (Fig. EV7). We find that in the absence of *Sox2* we see a dramatic loss of AQP5+ acinar cells and general tissue architecture following radiation, confirming that *Sox2* is essential for acinar cell replacement following IR injury. In addition, we find that other acinar cells do not switch on SOX2 in response to IR stress (Fig. EV7).

10. There are large numbers of Sox2+ cells in ex vivo explants - what is the level of cell proliferation? Does carbachol increase cell proliferation, or only differentiation of Sox2 progenitors? (Fig. EV4)

We see a comparable number of SOX2+ cells in the ex vivo cultures after 48 hours in culture as we do in freshly dissected tissue. In this assay we demonstrate an increase in the extent of lineage tracing (i.e. GFP positive cells that have arisen from a SOX2+ cell) with the addition of CCh, demonstrating that cholinergic stimulation increases SOX2-mediated cell replacement ex vivo. We also find a substantial increase in SOX2+ cell proliferation upon CCh treatment and we have now included this data here (Fig. EV6E) and added text as follows:

“This increase in GFP+ clones was associated with an increase in cell proliferation (Ki67+ cells) with CCh treatment (Fig. EV6E).”

This is consistent with our new in vivo data where we show that mice treated with cholinergic agonist, pilocarpine, exhibit an increase in SOX2+ cell proliferation, compared to saline control (new data shown in Fig. 4). Thus, we conclude that cholinergic stimulation induces SOX2+ cell proliferation, and subsequent repopulation of acinar cells during homeostasis and following injury.

11. Were the ex vivo experiments shown in Fig.5B-D done with healthy non-IR tissue or with IR-

treated tissue? This is not clearly specified. The data show that culture with parasympathetic nerves can increase Sox2, and other markers - does it increase proliferation? No conclusions regarding proliferation (Fig. 5C) are stated. Were explants only from SM or also from parotid samples?

We apologize for the confusion surrounding this assay. The human tissue used here was isolated from healthy non-IR patients. We demonstrate that PS nerves increase the numbers of Ki67+ proliferating cells in the epithelium (marked by E-cadherin) (Fig. 6C and quantified in EV8C) and this is discussed in the text as follows:

“In addition, epithelial cell proliferation, marked by the presence of Ki67+ cells, was increased in the presence of the nerves (Fig. 6C).”

Explants were isolated from SMG only (Fig. 5B-D) and SMG or PG (Fig. 5E) and this has been clarified in the methods text.

12. The authors have included an unrelated experiment in the manuscript showing short-term (14 day) lineage tracing of cKit+ cells in the mouse. Based on the rate of cell turnover in the salivary gland, this is an incomplete experiment, and should be removed - or expanded to include longer tracing times. These data do not contribute to the Sox2 story.

The thank the reviewer for this comment and have clarified why this data is important as well as provided longer lineage tracing times (14 days and 6 months). Current dogma in the field proposes that KIT+ cells are stem/progenitor cells in the salivary gland capable of regenerating secretory tissue - including acinar cells. However, to date no lineage tracing studies have been published demonstrating which cell types these KIT+ cells contribute to in the murine SG. Thus, for the purposes of our study and for the field in general we think it's very important to show that KIT+ cells do not contribute to acinar cell replacement in contrast to SOX2+ cells. To make this analysis more robust we have added an additional time point at 6 months demonstrating that even after 6 months of tracing we still see no contribution to acinar cell replacement by KIT+ cells.

We have updated the text accordingly as follows:

“Given KIT+ cells, which reside primarily in the intercalated ducts of the SLG and SMG (Andreadis et al, 2006; Nelson et al, 2013), have previously been proposed to give rise to acinar cells in adult tissue (Lombaert et al, 2008; Nanduri et al, 2014; Nanduri et al, 2013; Pringle et al, 2016b), we genetically traced these cells using the *Kit*^{CreERT2} promoter crossed to the *Rosa26*^{mTmG} reporter at 6 wks of age. However, no KIT+ cell-derived acinar cells (i.e. double positive for AQP5 and GFP) were evident in either the SLG or SMG after 14 days or 6 months after induction (Fig. EV1F). Instead, KIT+ cells contributed exclusively to the intercalated ducts in the SLG (as can be observed by co-staining with the intercalated duct marker KRT8) and intercalated and larger ducts in the SMG. Thus, these data indicate that KIT+ cells are progenitors for the ductal and SOX2+ cells for the acinar lineage. Given KRT5/14 (KRT5 and KRT14 are co-expressed) also mark a progenitor population in the adult salivary glands that give rise to larger ducts and myoepithelial cells but not acinar or intercalated duct cells (Kwak et al, 2016), we conclude that there are at least 3 progenitor cell populations that contribute to distinct epithelial compartments in the adult SLG.”

Additional points:

13. Were the 2x5Gy doses sequential, different days? Whole head/neck - where was dose rate of 5 Gy calculated? the surface of the mouse? Or the dose midway through the neck?

The head and neck region of the mice were irradiated with a bilateral dose of 5Gy (i.e. 5 Gy dose on each side of the mouse) at a dose rate of 167 Rads/minute for 2.59 min (5 Gy), for a total dose of 10 Gy. The mice were irradiated by being placed into a Shepherd Mark I Irradiator (JL Shepherd & Associates). The body and the very anterior part of the mouth (the snout) was shielded from radiation using lead blocks. Thus, we irradiated the head and neck regions only. Since the irradiator uses a stationary dose rate, the position within the irradiator where the mice are placed determines how much dose it receives. See <http://www.jlshepherd.com/> for more information.

We have clarified this in the methods as follows:

“The mice were irradiated using a ^{137}Cs source by being placed into a Shepherd Mark-I-68A ^{137}Cs Irradiator (JL Shepherd & Associates). Two lead, positioned 1.5 cm apart, were used to shield the body and the very anterior part of the mouth (the snout) of the mice and expose only the neck and part of the head. The 1.5 cm opening was centered at position 3 (20 cm from the ^{137}Cs source, 15.5 cm from the width of the irradiator cavity). Mice were exposed to 2 doses of 5 Gy at a dose rate of 167 Rads/minute for 2.59 min (one of each side of the head, bilateral and sequential but on the same day) for a total dose of 10 Gy, to irradiate the salivary glands. This does was calculated by Isodose plot mapping (dose distribution), provided by the manufacturer and EBT films (Brady, Toncheva G Fau - Dewhirst et al.) were used to localize the 100% region of exposure for mouse placement.”

14. Is the GFP shown in most figures imaging direct fluorescence, or obtained through staining with GFP antibody? This should be included in Methods section.

We imaged endogenous GFP signal in OCT embedded tissue/cryosections but due to processing for paraffin embedding, which quenches GFP, we needed to use a GFP antibody for wax sections (such as the staining in Fig. 1C; Chicken anti-GFP; 1:500, Aves Labs, GFP-1020). We have clarified this in the methods.

15. Not specified: Are values presented in Fig.5E averaged from samples incubated with 2 concentrations of CCh (100 and 200 nM)?

We apologize for the confusion. We treated with a dose of CCh ranging from 0nM to 200nM but the results shown here are 200nM. We have clarified this in the methods and legends.

16. Reference to Fig. EV5A on page 10, should be corrected to EV4.

We apologize for this oversight and we have corrected this in the text.

17. Labeling of all figures should be modified to indicate that GFP is derived from Sox2CreER mTmG and not from Sox2CreERGFP (which suggests a GFP insert linked to the CRE).

We have modified the labelling of figures in line with the reviewer's suggestion.

Referee #3 (Comments on Novelty/Model System):

a. Human tissue is used to identify Sox2 cells and mouse models are used to support this data. The gland examined in mouse is not the same gland as the human and this difference is not addressed.

We apologize to the reviewer for this. SOX2 is restricted to the adult SLG gland only in the mouse. SOX2 is absent in the SMG and PG. We have now included images in Figure 1 and Figure EV1 to demonstrate this. In contrast, SOX2 is expressed in all three of the major human glands (SMG, SLG, PG) and we have included images for all three in Figure 1 and as such the murine SLG provides a means to model human salivary glands. We have now clarified this in the text as follows:

“We found SOX2 to be expressed by a subset of acinar cells in all three of the major adult human salivary glands (Fig. 1A, submandibular gland (SMG), sublingual gland (SLG), parotid gland (PG)). In the mouse, SOX2 protein was restricted to the adult murine SLG (absent from the SMG and PG, Fig. 1B and Fig. EV1A) where it was expressed by undifferentiated aquaporin (AQP)5+ mucin (MUC)19-negative acinar cells (21% ± 4% of all AQP5+ acinar cells; Fig. 1C and D).”

b. Additionally, IR model in mouse does not reflect human findings and differences are not adequately addressed.

We thank the reviewer for this comment. In our studies we have now shown that, contrary to the current dogma, murine salivary glands have regenerative capacity following IR via targeting SOX2+ cells and maintaining cholinergic innervation. However, in human patients, at the time points we have assessed (>2 years following IR) there is a lack of regeneration of parasympathetic nerves and a subsequent loss of SOX2+ cells. We believe that by understanding the differences between

regeneration in our murine models and in human patients will direct potential therapeutic approaches to regenerate human glandular tissue following radiation injury. We have clarified this in the text in the Discussion as follows:

“Previous studies have utilized the murine salivary gland as a model of radiation-induced degeneration (Coppes et al. 2002; Coppes et al. 2001; Zeilstra et al. 2000). These investigations have been based on the assumption that regeneration is impaired after moderate to high doses of radiation, a hypothesis supported by the reduced saliva flow measured in animals receiving radiation (Redman, 2008). However, to date there has been no *in vivo* analysis of cell replacement after radiation. Our data indicate that murine acinar cells are highly regenerative, at least in the first 30 days after radiation exposure, and are capable of repopulating the acini similar to uninjured controls. It is clear, however, that this regenerative capacity cannot be sustained for the long term as degeneration/senescence in murine salivary glands occurs 3-6 months after radiation (Marmary et al. 2016; Urek et al. 2005). As such, it is likely that the regenerative capacity of SOX2+ cells does fail eventually and further analysis is required to discern the cause. It also remains to be determined whether the human salivary gland can regenerate in the days/months after therapeutic radiation and if this regenerative capacity fails in the long term due to absence of SOX2+ cells in combination with parasympathetic nerves. Indeed, a time-course analyzing changes in salivary glands from patients is required to understand how these organs are affected in the short- and long-term. However, our results suggest that targeting these stem cells and their innervating nerves to control and sustain tissue regeneration in response to radiation damage may provide a means of maintaining/repairing tissue for the long term.”

Referee #3 (Remarks):

In this manuscript, the authors have identified a Sox2-positive population of cells in the submandibular salivary glands of humans that is capable of generating acini in mouse sublingual glands. During homeostasis, loss of Sox2 leads to decreased AQP5 and increased ducts in the sublingual gland of mice. The authors demonstrate that this Sox2-positive population is positively maintained by parasympathetic innervation in the mouse and imply that this mechanism may be conserved in humans. The authors suggest that human glands do not regenerate due to loss of Sox2+ progenitor cells and the nerves that regulate them. Reintroduction of innervation to the human tissues with CCh stimulation in culture is shown here as a novel co-culture assay. The findings are interesting and likely have relevance to human disease; however, there are significant issues that should be addressed.

Major points:

1. Only the sublingual gland (SL) is examined in mice in this study and so that authors should explicitly state that the study uses SL in the abstract and elsewhere in the manuscript.
2. It is unclear why SL mouse glands are compared with submandibular and parotid but not SL human.

We apologize for the inconsistency and lack of information. SOX2 is restricted to the adult SLG only in the mouse and is absent in the SMG and PG (see response to earlier point). However, the adult human SMG, SLG and PG all express SOX2 and as such the mouse SLG provides a good model for all three major human salivary glands. The fact that human SMG or PG is used in all *ex vivo* assays is due to the availability of these tissues. Surgery and dissection to remove human SLG is extremely rare and as such we have very limited access to fresh tissue, whereas fresh human SMG or PG is more readily available for our studies.

3. Sox2 is a known potency marker. Are these cells positive for other known potency markers- eg. Oct4, Klf4, or Nanog?

We agree with the reviewer that this is an interesting question. In order to address this we have demonstrated by immunostaining of adult SLG (using mouse ES cells as a positive control and the 44cre6 Nanog null line and ZHBTC4.1 Oct4 conditional null line as negative controls) that SOX2+ cells do not express either OCT4 or NANOG (see Rebuttal Fig. 1A and B at end of letter).

4. Since neither Sox2+ cells are eliminated with genetic manipulation of mice, nor are acini, it is possible that there are other mechanisms for regulation of repopulation of the acinar population, including as the authors state does not occur, control through Mist1. Since the authors have not tested this directly, and have not costained for this marker, their conclusion that Sox2+ cells are the only progenitor cells and that the Mist1 population does not contribute to tissue restoration is a gross overstatement. Interestingly, the authors report increased Mist1 transcripts in human tissue culture co-culture assays and in assays with CCh stimulation and should comment on this.

We apologize for the confusion. We do genetically ablate the SOX2+ cells using diphtheria toxin (*Sox2CreERT2;Rosa26DTA*). This results in profound loss of the acinar compartment indicating that other cells (MIST1+SOX2-) do not repopulate the tissue. We have clarified these experiments in the text.

“... we investigated the requirement of SOX2 and SOX2+ cells in SLG maintenance and repair by genetically removing *Sox2* in SOX2+ cells using *Sox2^{CreERT2}*; *Sox2^{fl/fl}* mice (Fig. 2A, C) or ablated SOX2+ cells using diphtheria toxin (DTA) expressed under the control of the inducible *Sox2* promoter (*Sox2^{CreERT2}*; *Rosa26^{DTA}*; Fig. 2B, D). In the latter assay, SOX2+ cells undergo cell death in response to intracellular accumulation of DTA. Ablation of *Sox2* from SOX2+ cells or elimination of SOX2+ cells via DTA severely depleted SOX2+ and AQP5+ cells but not KRT8+ ductal cells indicating *Sox2* and SOX2+ cells were necessary for maintaining functional acini (Fig. 2A-D; efficiency of *Sox2* or SOX2+ cell ablation is shown in Fig. EV2A). In the absence of *Sox2*, acinar but not ductal cells exited the cell cycle, as shown by the decrease in cyclin D1 (CCND1)+ acinar cells (Fig. EV2D, arrowheads indicate CCND1+ cells and dotted white lines highlight ductal cells). Furthermore, ablation of SOX2+ cells resulted in few remaining acini by 8 days (Fig. 2B, D, Fig. EV2A), as shown by large regions of the ductal network completely devoid of AQP5+ cells (ducts are marked by dotted lines or KRT8 in Fig. 2B). To exclude the possibility that tissue degeneration was solely due to destabilization of the tissue rather than loss of acinar cell replacement, we examined SLG after a short-term ablation. As shown in Fig. EV3A and 3B, at day 4 or 5 (3 or 4 days of tamoxifen treatment), few SOX2+ cells remained in the gland of both the *Sox2^{CreERT2}*; *Sox2^{fl/fl}* and *Sox2^{CreERT2}*; *Rosa26^{DTA}* SLG (Fig. EV3A, B) and *Sox2* transcripts were substantially reduced (Fig. EV3B). However, acini were present albeit disorganized and atrophic in appearance. Furthermore, we did not observe an increase in SOX2+ cells (or *Sox2* transcripts), indicating that SOX2 is not ectopically expressed in acinar cells in response to tissue damage.”

We were thrilled that *MIST1* was positively regulated in the human cultures. Given MIST1 is a regulator of the secretory program, increased expression may correlate with an increase in genes important for acinar function. As such, we have included a comment on the increase in MIST1 in the human co-cultures as follows:

“While surgical denervation does not adversely affect expression of *Mist1* (Fig. 3B, E) muscarinic stimulation is sufficient to increase *MIST1* in human cultures and suggests that although not required for acinar cell identity, acetylcholine/muscarinic signaling may act as a positive regulator of the secretory program.”

5. Fig EV1C In order to demonstrate that kit+ cells contribute to intercalated ducts, they should be identified with a marker.

6. Additionally, a lineage commitment marker for kit+ cells in the submandibular gland needs to be shown to make the statement that kit+ cells do or do not contribute to acinar cells in this gland.

We thank the reviewer for these constructive comments. We have now included images of KIT lineage traced cells (shown via GFP) with co-staining for AQP5 (acinar cells) and KRT8 (duct cells) (Fig. EV1F). We have clarified this in the text as follows:

“However, no KIT+ cell-derived acinar cells (i.e. double positive for AQP5 and GFP) were evident in either the SLG or SMG after 14 days or 6 months after induction (Fig. EV1F). Instead, KIT+ cells contributed exclusively to the intercalated ducts in the SLG (as can be observed by co-staining with the intercalated duct marker KRT8) and intercalated and larger ducts in the SMG. Thus, these data indicate that KIT+ cells are progenitors for the ductal and SOX2+ cells for the acinar lineage.”

7. Fig 2/FigEV2 Use of Sox2 fl/fl mouse does not appear to result in a severe lack of Sox2GFP area (Fig2A) as the graph in Fig 2C suggests

We apologize to the reviewer that this result is not clear. Rather than “a severe lack of Sox2GFP area” in Figure 2 with a loss of *Sox2* we report a “severe depletion SOX2+ cells”, which we have shown in Fig. EV2 (SOX2 staining) and this is quantified in 2C (right graph). The GFP channel (in red in the lower panels in Fig. 2A) demonstrated Cre-mediated deletion of *Sox2* and confirms that *Sox2* has been deleted in the *Sox2^{CreERT2}; Sox2^{fl/fl}* and not in the wild-type littermates.

Nor does there seem to be any significant change in innervation (Fig EV2).

The reviewer is correct and we do not report any difference in innervation in the Sox2 flox study and this is shown in EV2B and quantified in EV2C. This data supports our hypothesis that the alteration in morphology is due to the loss of SOX2 and not of the nerve supply (which we show is important for maintaining SOX2 and acinar cells. The primary conclusion in this assay is that even in the presence of innervation the loss of SOX2+ cells leads to a loss of functional acinar cells.

Use of the Sox2 DTA mouse only, however, appears to result in a phenotype.

We apologize for the confusion surrounding this figure. The phenotype (i.e. loss of acini) is more extreme in the *Sox2^{CreERT2}; Rosa26^{DTA}* model, but is consistent with the *Sox2^{fl/fl}*, where we also see a striking loss of acini. We see a loss of AQP5+ acini in both conditions (see Fig. 2A and B, AQP5 staining; red) but the DTA model results in 20% of cells being killed directly. This means that functional acinar cells are not being replaced under both conditions but the assays are very different. We have clarified this in the text as follows:

“Furthermore, ablation of SOX2+ cells resulted in few remaining acini by 8 days, as shown by large regions of the ductal network completely devoid of AQP5+ cells (ducts are marked by dotted lines or KRT8 in Fig. 2B, D, Fig EV2A).”

Fig EV2 Sox2 staining appears to be reduced in the Sox2 fl/fl mouse and not the Sox2 DTA mouse images. Please quantify the Sox2-positive nuclei in these mice to indicate loss of Sox2.

We thank the reviewer for this comment and we have re-arranged Figures 2 and EV2 to make the data clearer. In EV2A we demonstrate that there are abundant SOX2+ nuclei present in wild-type tissue (green nuclei, left panel) but in *Sox2^{CreERT2}; Sox2^{fl/fl}* or *Sox2^{CreERT2}; Rosa26^{DTA}* tissue there are few to no SOX2+ nuclei remaining (middle and right panels and magnified view of *Sox2^{CreERT2}; Rosa26^{DTA}*). The green staining that is apparent is punctate and not nuclear and thus not true SOX2+ nuclear staining. We have also included quantification of SOX2+ cells for both the *Sox2^{CreERT2}; Sox2^{fl/fl}* and *Sox2^{CreERT2}; Rosa26^{DTA}* models (Fig. 2C,D).

We have also kept the colours consistent in panels 2A and B, showing that with both the loss of *Sox2* (*Sox2^{CreERT2}; Sox2^{fl/fl}*) or SOX2+ cells (*Sox2^{CreERT2}; Rosa26^{DTA}*) AQP5+ cells (shown in green) are lost. In addition, we have shown that in both KRT8+ ducts remain intact (lower panels, in green).

8. It is troubling throughout the figures that only three mice are included in calculations and yet statistics are shown. Is this because three samples from three mice are quantified to make n = 9? For example, Fig 2C-D It appears that 5 mice of each genotype were obtained and yet only 3 sections from each of 3 mice were used for quantification.

We apologize to the reviewer for this confusion. Due to the limited constraints of breeding female mice of the correct genotype for this analysis the experiment was carried out using an n=3 for both the *Sox2^{CreERT2}; Sox2^{fl/fl}* and *Sox2^{CreERT2}; Rosa26^{DTA}* models. We have corrected this oversight at one place in the figure legend where we wrongly stated n=5. The quantification results we show were performed on 3 fluorescent sections of each SLG from 3 mice per genotype as stated in the legend for Fig. 2. The data analysed was averaged per mouse and then averaged per genotype. While we appreciate that n=3 is a low number for an in vivo study the amount of variation in the data we show is so minimal that we are confident in our conclusions that a loss of *Sox2* or SOX2+ cells consistently leads to a loss of AQP5+ acinar cells. Specifically, the signal to noise ratio is very low,

likely because an inbred background strain has been used, meaning that meaningful conclusions can be drawn from smaller group sizes.

9. Fig3 Staining for decreased Muc19 is reported in the text but is not shown in the figure.

We apologize to the reviewer for the confusion, here we report decreased *Muc19* at the transcriptional level (i.e. RNA, based on qPCR data and shown in Fig. 3B).

10. Fig 3B Since changes in *Mist1* were observed with human tissue coculture and CCh stimulation in Fig5, please also report changes (if any) in *Mist1* in this figure with transection.

We thank the reviewer for this comment and have added in data showing MIST1+ staining, quantification of MIST1+ cells and *Mist1* gene expression in our denervation experiments. Perhaps surprisingly we find no change in MIST1/*Mist1* following denervation. This suggests that while acinar cells themselves (as marked by MIST1) are not lost 7 days after denervation markers of acinar function, such as AQP5 and *Muc19*, are affected. We have addressed this in the text as follows:

“Interestingly, transcript and protein levels of MIST1 were unchanged following denervation (Fig. 3B, E, F), suggesting that while functional markers of acinar cells are disrupted in the absence of innervation, acinar cell identity is not adversely affected.”

11. It should be explained why *Chrm1* and *Chrm3* have not changed after 7 days if innervation has been removed.

It is possible that compensatory mechanisms may maintain muscarinic receptors at this time point. This has been discussed in the text as follows:

“We did not observe a concurrent loss of the cholinergic muscarinic receptors *Chrm1* and *Chrm3* transcripts (Fig. 3B, red bars) however, it is possible that in the absence of parasympathetic innervation a compensatory mechanism may maintain *Chrm1* and *Chrm3* transcription”

Additionally, it’s unclear why *Tubb3* and *Vip* have increased beyond baseline levels.

We were surprised to find this increase in *Tubb3* and *Vip* as well and propose that this is due to hyperinnervation of the tissue 30 days following injury, possibly due to the release of neurotrophic factors. Investigation of this outcome is the focus of a new manuscript. We have clarified our findings in the text as follows:

“Surprisingly, we found elevated expression of these neuronal genes at day 30, suggestive of hyperinnervation, in response to the original injury.”

12. In Fig 3C, the authors report a selective reduction in parasympathetic innervation. To make the conclusion, a parasympathetic nerve marker should be used to demonstrate a decrease with a sympathetic marker that does not decrease. *Tubb3* used in this figure marks all nerves and is not a selective marker for parasympathetic innervation and does not appear to be decreased. Use of a more conspicuous color or single channel image and/or quantitative data are required.

Immunostaining shown in Fig. 3C and Fig. EV3C demonstrates that TUBB3+ nerves are reduced 7 days following nerve transection (compare left panel with right panel, original Fig. 3C is now Fig. EV3C). We now show immunostaining for GFRa2, a marker of parasympathetic nerves in Figure 3C, clearly showing reduction of this marker. We also show tyrosine hydroxylase (TH)+ sympathetic nerves in Fig. EV4. The SLG is innervated poorly by sympathetic nerves which we do find to be reduced following chorda tympani transection. As such, we cannot rule out that some of the effects we see following denervation are due to a loss of sympathetic nerves, but since the SLG is heavily innervated by parasympathetic nerves and to a far lesser extent by sympathetic nerves, it is reasonable to propose that parasympathetic signalling is responsible. Indeed, studies in humans using botulinum toxin, which directly inhibits parasympathetic function, results in tissue atrophy, further supporting the major role of parasympathetic nerves (Teymoortash et al. Intraglandular

application of botulinum toxin leads to structural and functional changes in rat acinar cells. *Br J Pharmacol.* 2007;152:161-7.)

13. Previously published literature in multiple organs, including salivary, indicates that IR induces tissue damage. The authors should include a marker to demonstrate efficacy of the IR in inducing a "genotoxic effect" - perhaps increased levels of proteins involved in DNA repair pathways or other previously reported changes characterized with IR.

We thank the reviewer for this comment. We have now included much more characterization of the effect of IR on the SLG (see fig. EV6), and included multiple time points following IR damage. Within this figure we have included markers of cell cycle and cell death to show that expression of the pro-apoptotic gene *Bax* and the cell cycle/DNA repair gene *Cdkn1a* (p21) increase in the early days following IR. In addition, the cell proliferation marker *Mki67* is reduced following IR. We have clarified this in the text as follows:

"Similar to previous studies in salivary glands (Avila et al, 2009), we found a 10 Gy dose induces DNA damage and cell cycle arrest as well as reduces cell proliferation in the SLG in the first day following IR, as shown by a substantial increase in the pro-apoptotic gene *Bax* and the cell cycle inhibitor *Cdkn1a* (p21) (Fig. EV6A) and a reduction in transcript levels of the cell proliferation marker *Mki67* (Fig. EV6A)."

14. Fig 4E To show that Sox2 is increased with IR at 48 hours in culture after IR, please quantify fluorescent areas for GFP and show representative cropped images of fluorescent areas.

We show two magnifications of images in this panel to allow the reader to see the gross morphology of the tissue (lower magnification, top panels) and the extent of GFP clones (as representative images of higher magnification, lower panels). We have now quantified the GFP+ cells as suggested by the reviewer and show this in the graph in Fig. 4F.

15. In Fig 5A the change in Sox2 is not significant ($p < 0.05$) and should be reported as a trend.

We apologize for this oversight, we have changed the wording of the text to reflect this.

16. To suggest decreased parasympathetic nerve function in human tissues, decreased levels of markers for parasympathetic nerve function should be shown as for the mouse model (eg. *Vip*, *Vacht*). As the authors state, Chrm markers are also present on acini and acini reported to be reduced, decreased *Chrm1* and decreased *Chrm3* may be due to decreased epithelium.

We thank the reviewer for this comment. As previously reported in Knox, et al. 2013. IR in human patients results in a loss of parasympathetic but not sympathetic innervation. We have added transcriptional profiling data for *TUBB3*, *VIP*, *GFRA2* and *TH* in to Fig. EV8. While we do not see a reduction in *VIP* (likely due to *VIP* being expressed by immune cells (Ganea et al. The neuropeptide *VIP*: direct effects on immune cells and involvement in inflammatory and autoimmune diseases. 2014. *Acta Physiol (Oxf)*. 213(2): 442-52) we do see a significant reduction in expression of *GFRA2*, a marker of parasympathetic nerves not expressed by immune cells (see also Knox, et al. 2013). Conversely, we do not see a change in the expression of the pan-neuronal marker *TUBB3* or the sympathetic nerve marker tyrosine hydroxylase (*TH*). Thus, we are confident in our conclusion, as previously reported (Knox, et al. (2013)) that IR in humans results in a loss of parasympathetic nerves.

17. Fig5 C Ki67 staining appears higher in the mesenchyme only co-culture than with nerve co-culture, and ECAD appears to be reduced. Since ECAD levels appear to be significantly different in this assay, and the genes examined are epithelial, it's important to normalize to epithelium in addition to mesenchyme to account for this difference. Alternatively, demonstrate that there is no significant difference in the amount of epithelium present and include more representative images.

We agree with the reviewer that ECAD expression is different between the explants cultured with mesenchyme only versus those cultured with mesenchyme and PSG. This is likely due to the fact that following 7 days in culture the explants are surviving better in the presence of nerves and as such are better expressing markers of healthy epithelia than the mesenchyme alone. In addition, all

assays were set up with equivalent amounts of epithelium and mesenchyme. ECAD expression in such an assay doesn't necessarily equate to the amount of epithelia in the assay, merely epithelial adhesion. Thus, we feel that ECAD expression is not an appropriate control to normalise to.

18. Fig5 D KIT and EGFR results indicate one sample of two at baseline, not increased. Myoepithelial/ductal markers also appear to be at or near baseline. Please correct text to reflect results reported.

We apologize for the confusion here. We have corrected the text to reflect this.

19. Fig5 E/Fig EV5B Please clarify if human samples from different glands are averaged and if Fig 5/Fig EV5 are parts of the same data set as Fig. 5D. Methods/legend/ and results are unclear for Fig 5E/D.

We apologize for the confusion. Fig. 5E and 5D (now 6D and 6E) are completely separate experiments. The glands used for 6D came from 2 separate individuals (n=2) and the glands used in 6E came from 4 separate individuals (n=4) and are unrelated to the experiments shown in 6D. We have clarified this in the text as follows:

“Fig. 5D, data normalized to mesenchyme only, n=2 separate individuals”
 “Culture of patient-derived human tissue from 4 separate individuals (n=4)...”
 “(Fig. 5E, n=4, individual datasets shown in Figure EV5B).”

Fig EV5B indicates variability in the dataset that should be discussed. Additionally, if Fig. 5 is a summation of EV5 results, please indicate this.

We thank the reviewer for pointing this out and we have clarified this in the text as follows:

“(Fig. 5E, n=4, individual datasets shown in Figure EV5B)”
 “The variability in response between the 4 patient-derived samples is likely due to biological diversity between human patients, differences in the type of gland sourced (SMG and PG used) and the age of the patient (age of donor ranges from 30 to 78 years). However, in all cases we observed an increase in SOX2 and a number of acinar markers in the presence of CCh.”

Minor points:

1. DAPI and other markers shown blue are written in too dark of a color to show up in all of the figures. Also the use of blue in the 3 color overlays without single panel images makes it very difficult to discern the blue color. The authors should consider a color closer to cyan than blue.

We thank the reviewer for this comment. However, in all situations the structures shown in blue are of the least relevance to the data interpretation (i.e. nuclei, epithelium; used for markers of structure), and hence shown in blue rather than a more distinct colour which we reserve for the markers of substantial interest (red/green). We have stuck to primary colours as they are very discernible to the human eye and the use of colours such as cyan for nuclei detracts the attention away from similar colours, such as green, when used in a merge. However, we have increased the brightness of the text so that all labels in blue are clearly legible.

2. Kwak, Alston et al., 2016 does not use K5 as a marker in this study as incorrectly stated bottom of page 6/top of page 7

We apologize for this, we have cited this paper as it shows that KRT14 cells give rise to large ducts and myoepithelial cells. Since KRT5 and KRT14 mark the same cells we postulate that KRT5+ cells also give rise to the same structures, however lineage tracing experiments have not been reported for KRT5. We have clarified this in the text as follows:

“Given KRT5/14+ cells (KRT5 and KRT14 overlap and mark the same progenitor population in the adult salivary glands) have been shown to exclusively give rise to larger duct and myoepithelial cells and not acinar cells (Kwak, et al. 2016)”

3. Fig. 2. DAPI in bottom right panel of Fig 2A and Fig 2B appears to be brighter than other images. Are the exposures matched?

We image all sections using the same confocal settings and do not adjust the brightness of an image without altering all images at the same time. The exposures for these images are thus matched and comparable. The fact that the nuclei staining looks brighter in each of the right hand panels is likely due to the fact that in both of these situations acinar structures have been lost and subsequently there is less structure surrounding the nuclei so they appear brighter to the eye.

4. Fig EV3A Why is KRT5 shown and not referred to? What is the relevance of this marker to the model? Need to clarify

We apologize for the omission here and have clarified the reason for looking at KRT5 in better detail in the text as follows:

“Furthermore, KRT5+ cells, progenitors in developing SMG/SLG (Knox et al, 2010) that are maintained by parasympathetic nerves (Knox et al, 2010) were unaffected by denervation (Fig. EV4D and Fig. 3F; for transcript *Krt5* expression see Fig. 3B).”

5. Fig EV4 is reported incorrectly in the text as EV5A.

We apologize for this mistake and thank the reviewer for bringing it to our attention. We have corrected this in the text.

Rebuttal Figure 1. The pluripotency factors NANOG and OCT4 are not expressed in SOX2+ cells in the murine SLG. (A) Adult SLG was immunostained for SOX2, NANOG, epithelia and nuclei. (B) Adult SLG was immunostained for SOX2, OCT4, epithelia and nuclei.

3rd Editorial Decision

04 December 2017

Thank you for the submission of your revised manuscript to EMBO Molecular Medicine. We have now received the enclosed reports from the referees that were asked to re-assess it. As you will see the reviewers are now globally supportive and I am pleased to inform you that we will be able to accept your manuscript pending the following final amendments:

1) Please address the minor text change commented by referee 2 and provide a point-by-point response to my queries and to the referee's. Make sure to reflect the referees' comments in your final article.

***** Reviewer's comments *****

Referee #1 (Remarks for Author):

The authors have addressed my concerns, so I recommend the manuscript to be accepted for publication.

Referee #2 (Comments on Novelty/Model System for Author):

The authors have responded to most of the reviewer's comments and critiques, although the quality of the supporting data are inconsistent, and require some attention. Specifically, data added to show long-term lineage tracing of cKit cells include high background, and lack of co-localization, such that they do not clearly prove what cell types are derived from cKit cells. Figure 4 is another example. Sox2 staining in Fig. 4a is poor and even the ecadherin staining does not look normal in this photo. In photos to show that Sox2+ cells are proliferating, it is impossible to see co-localization with the colors used. Furthermore, the major message of the manuscript is that Sox2 cells are required for regeneration, but there has been no attempt to directly look at regeneration. The message that this work relays is that Sox2 cells are important for acinar cell survival and expansion.

Referee #2 (Remarks for Author):

The authors should note that while Figure 1 does show Sox2 expression limited to acinar cells, other photos (especially Figure 5) have distinct evidence of Sox2 in the duct cells - although the authors have argued that Sox2 is acinar cell-specific. Fig. 2A) Please explain what is the red stain or antibody that is labeled as "cell membrane". Fig. 2: white outlines to show ducts aren't even aligned over the ducts. In Fig. 2B, the DTA model has entire acini of aquaporin5+ cells included in the outline of what is supposed to be ducts. Fig. 2EVA Sox2CreERT2/Rosa26 DTA photos - it is not at all clear what these photos are supposed to show - The green is supposed to mark Sox2 cells? Figure legend for EV3A incorrectly states that photo shows DTA tissue, when the photo is labeled as Sox2 fx/fx tissue.

The inability of Sox2 fx/fx SLG to maintain or regenerate Aqp5+ cells after IR is an interesting and convincing experiment. Why is it relegated to supplementary figures?

Referee #3 (Remarks for Author):

My concerns have been adequately addressed.

2nd Revision - authors' response

14 December 2017

(begins on next page)

Reviewer Reports

Referee #1 (Remarks for Author):

The authors have addressed my concerns, so I recommend the manuscript to be accepted for publication.

Referee #2 (Comments on Novelty/Model System for Author):

The authors have responded to most of the reviewer's comments and critiques, although the quality of the supporting data are inconsistent, and require some attention.

Specifically, data added to show long-term lineage tracing of cKit cells include high background, and lack of co-localization, such that they do not clearly prove what cell types are derived from cKit cells.

We thank the reviewer for this comment. We have updated the images to clearly demonstrate that lineage traced KIT+ cells (GFP+) are not AQP5+ but are KRT8+ (i.e. ductal, not acinar).

Figure 4 is another example. Sox2 staining in Fig. 4a is poor and even the ecadherin staining does not look normal in this photo.

We agree with the reviewer and have replaced this image with two images clearly showing expression of CHRM3 by SOX2+ cells.

In photos to show that Sox2+ cells are proliferating, it is impossible to see co-localization with the colors used.

We thank the reviewer for this comment. We have now included clearer images showing co-localization of SOX2 and Ki67 (double positive "yellow" cells).

Furthermore, the major message of the manuscript is that Sox2 cells are required for regeneration, but there has been no attempt to directly look at regeneration. The message that this work relays is that Sox2 cells are important for acinar cell survival and expansion.

We kindly disagree with the reviewer based on our new data showing no replenishment of AQP5+ acini after gamma radiation in glands in which Sox2 has been genetically ablated. In Figure 5D, we demonstrate using genetic lineage tracing that, following radiation, SOX2+ cells contribute to acinar cells in the SLG (imaged at 14 days post-radiation). We further show that 14 days after radiation acini express AQP5 protein at levels comparable to non-irradiated tissue, indicating successful regeneration of acini (Fig.5E, left panels). However, also in Figure 5E we show that in the absence of Sox2 the gland fails to restore AQP5+ cells after radiation injury (again imaged 14 days post-radiation). Based on this data, we conclude that SOX2+ cells are required for SLG regeneration following IR injury due to the inability of tissue in the absence of Sox2 to restore AQP5+ acinar cells.

Referee #2 (Remarks for Author):

The authors should note that while Figure 1 does show Sox2 expression limited to acinar cells, other photos (especially Figure 5) have distinct evidence of Sox2 in the duct cells - although the authors have argued that Sox2 is acinar cell-specific. Fig. 2A)

We show on multiple occasions in the manuscript that the duct cells do not express SOX2 (Figures 1, 3C, 5D, EV1E and EV2A). We also provide the reviewer with another image at the end of this letter showing that SOX2 is expressed by AQP5-positive but not AQP5-negative cells. If the reviewer is referring to untraced cells (i.e. those in blue) this demonstrates that there are SOX2+ cells that were not traced (likely due to recombination efficiency being below 100%), not that they are ductal cells.

Please explain what is the red stain or antibody that is labeled as "cell membrane".

We apologize for the confusion at not describing the cell membrane markers. In the images in Fig. 2A cell membrane refers to ECAD (left panel) or mGFP (right panel, membrane bound tomato), and in Fig.2B it refers to mT. We have updated the text accordingly.

Fig. 2: white outlines to show ducts aren't even aligned over the ducts. In Fig. 2B, the DTA model has entire acini of aquaporin5+ cells included in the outline of what is supposed to be ducts.

We thank the reviewer for noticing this error and have ensured that all dashed lines are perfectly aligned with the ductal structures they are denoting in Fig.2B.

Fig. 2EVA Sox2CreERT2/Rosa26 DTA photos - it is not at all clear what these photos are supposed to show - The green is supposed to mark Sox2 cells?

This figure shows that Sox2 and SOX2+ cells were efficiently ablated using the $Sox2^{CreERT2}; Sox2^{fl/fl}$ and $Sox2^{CreERT2}; Rosa26^{DTA}$ mice, respectively. The green staining in Fig. EV2A marks SOX2+ nuclei. In the magnified view of the $Sox2^{CreERT2}; Rosa26^{DTA}$ image demonstrates that any immunostaining (green) is not specific to SOX2+ nuclei and is likely debris.

We have clarified this in the main text and the Figure Legend for EV2.

Main text: "Ablation of Sox2 from SOX2+ cells or elimination of SOX2+ cells via DTA severely depleted SOX2+ and AQP5+ cells but not KRT8+ ductal cells indicating Sox2 and SOX2+ cells were necessary for maintaining functional acini (Fig. 2A-D; efficiency of Sox2 or SOX2+ cell ablation is shown in Fig. EV2A)."

Figure legend text: "**Figure EV2. Ablation of Sox2 or SOX2+ cells reduces acinar cell replacement despite the presence of nerves (A-C)** Sox2 or SOX2+ cells were ablated in SLG of $Sox2^{CreERT2}; Sox2^{fl/fl}; Rosa26^{mTmG/+}$ mice (**Figure 2A**; see schematic) or $Sox2^{CreERT2}; Rosa26^{DTA}; Rosa26^{mTmG/+}$ mice (**Figure 2B**; see schematic). (**A** and **B**) Sections of WT, $Sox2^{CreERT2}; Sox2^{fl/fl}$, and $Sox2^{CreERT2}; Rosa26^{DTA}$ SLG were immunostained for SOX2 or TUBB3. White arrowheads indicate SOX2+ cells. White dotted square is magnified in the image to the right to highlight that there are few SOX2+ cells remaining in tissue and that non-nuclear (green) immunostaining is suggestive of debris. Scale bar = 50µm."

Figure legend for EV3A incorrectly states that photo shows DTA tissue, when the photo is labeled as Sox2 fx/fx tissue.

We thank the reviewer for noticing this error. Panel EV3B shows $Sox2^{CreERT2}; Rosa26^{DTA}$ SLG while EV3A shows $Sox2^{CreERT2}; Sox2^{fl/fl}$. We have corrected this mistake in the Figure legend.

The inability of Sox2 fx/fx SLG to maintain or regenerate Aqp5+ cells after IR is an interesting and convincing experiment. Why is it relegated to supplementary figures?

We agree with the reviewer about the importance of this figure and have subsequently moved it to be part of the main manuscript as Figure 5E.

Referee #3 (Remarks for Author):

My concerns have been adequately addressed.

Figure for reviewer #2. Image shows SOX2 is expressed by AQP5-positive cells and not AQP-negative cells

Corresponding Author Name: Sarah Knox

Journal Submitted to: EMBO Molecule Medicine

Manuscript Number: EMM-2017-08051-V2